# DECOR: Learning to Decompose and Collaborate in Deep Search via Multi-Agent Reinforcement Learning

**Ruiqing Chen** [1] **Zekun Zhang** [1] **Gong-Duo Zhang** [1] **Lihong Gu** [1] **Lin Zhou** [1]

## Abstract

Monolithic agents in deep search often suffer from "cognitive overload," while existing multi-agent approaches mostly rely on frozen models that cannot learn from collaboration failures. To bridge this gap, we propose **DECOR** (**DE**compose and **CO**llaborate via **R**ole-specialized agents), a framework formulating deep search as a Multi-Agent Reinforcement Learning (MARL) problem. DECOR functionally decomposes the task into three specialized roles: a *Planner* to navigate, a *Filter* to curate a noise-reduced memory, and an *Answerer* for synthesis. Unlike training-free orchestration, we jointly optimize these agents using a hybrid reward strategy that harmonizes role-specific intrinsic feedback with team-level outcome signals. Experiments on seven benchmarks show that DECOR significantly outperforms strong monolithic baselines, demonstrating the necessity of learning-based functional decomposition in handling cognitive overload.

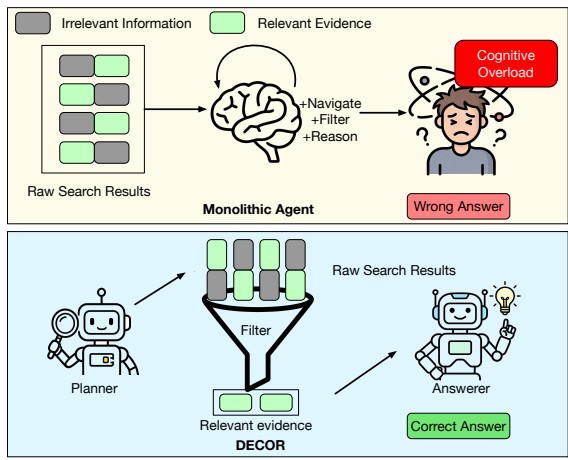

*Figure 1.* Comparison between a monolithic agent and the proposed DECOR framework. (*Top*) A monolithic agent suffers from **cognitive overload** as it attempts to simultaneously navigate, filter, and reason over raw search results, often leading to hallucinations. (*Bottom*) **DECOR** functionally decomposes the task into specialized roles. By introducing a dedicated *Filter* agent to curate a noise-reduced memory, DECOR effectively shields the *Answerer* from irrelevant context, enabling it to focus purely on logical deduction.

## 1. Introduction

Large Language Models (LLMs) now tackle complex reasoning through iterative interactions with external environments. Central to this is *deep search*: autonomously planning retrieval, distilling information, and synthesizing answers. While prompting strategies like ReAct (Yao et al., 2022) attempts to encapsulate these capabilities within a single context, they face a fundamental bottleneck: **cognitive overload** (Xu et al., 2024; Liu et al., 2024). As retrieved contexts accumulate, a monolithic agent's attention becomes diluted, forcing it to simultaneously act as a navigator, a noise filter, and a reasoner. This multi-tasking burden often leads to role confusion, where the model becomes distracted by irrelevant results or loses track of its reasoning chain,

ultimately hallucinating answers despite having access to correct information.

To mitigate the burden on monolithic models, the community has shifted toward multi-agent architectures that decompose complex tasks into sub-routines. However, most existing multi-agent approaches function primarily as *inference-time optimizations*. They rely on hand-crafted Standard Operating Procedures (SOPs) or dynamic architecture search (e.g., selecting effective agents) to orchestrate frozen LLMs. While these systems introduce a structural division of labor, they typically lack a mechanism to *fine-tune the intrinsic policies* of the agents based on interaction history. For instance, a Planner in a static framework cannot learn to refine its queries solely based on the Answerer's downstream confusion. Consequently, the potential of collaborative intelligence remains bottlenecked by the static capabilities of the base models and the inability to solve the *multi-agent credit assignment problem*—determining which specific agent's action led to a success or failure.

[1]Ant Group, Hangzhou, China. Correspondence to: Ruiqing Chen <chenruiqing.crq@antgroup.com>.

*Proceedings of the $43^{rd}$ International Conference on Machine Learning*, Seoul, South Korea. PMLR 306, 2026. Copyright 2026 by the author(s).

In this paper, we propose **DECOR** (**DE**compose and **CO**llaborate via **R**ole-specialized agents), a framework that addresses these limitations by formulating deep search as a collaborative MARL problem. While general multi-agent RL frameworks exist, DECOR is, to the best of our knowledge, the first to apply heterogeneous policy optimization specifically tailored for the deep search pipeline. We assign distinct roles to three specialized agents: a *Planner* responsible for navigating the search space, a *Filter* dedicated to curating a high-signal memory pool by pruning irrelevant contexts, and an *Answerer* that synthesizes the final response. This design explicitly addresses the cognitive overload problem: by offloading information filtering to a dedicated agent, the Answerer is shielded from noise, allowing it to focus purely on logical deduction.

A key innovation of DECOR is its solution to the credit assignment challenge in heterogeneous teams. Training such a team is non-trivial: standard RL methods that propagate a single scalar reward fail to disentangle whether a failure was caused by a poor query (Planner), a missed piece of evidence (Filter), or flawed logic (Answerer). To resolve this, we introduce a **hybrid reward strategy** that combines role-specific feedback with team-level outcome signals. Utilizing an LLM-as-a-Judge, we provide dense, intrinsic rewards for intermediate actions while jointly optimizing all agents to maximize the final answer accuracy. This allows DECOR to evolve from a rigid chain into a dynamic team where the Planner and Filter actively learn to reduce the reasoning burden on the Answerer. Our contributions are summarized as follows:

- We propose DECOR, a modular framework that functionally decouples information seeking from answer synthesis. By introducing a dedicated *Filter* agent to curate retrieved contexts, our architecture effectively shields the reasoning module from noise, mitigating the cognitive overload inherent in monolithic agents.

- We develop a collaborative MARL training paradigm driven by a hybrid reward strategy. By harmonizing team-level outcome signals with dense, role-specific intrinsic rewards, DECOR effectively resolves the credit assignment problem and enables joint optimization of heterogeneous agents.

- Extensive experiments on seven mainstream benchmarks demonstrate that DECOR significantly outperforms baselines, validating the necessity of learning-based functional decomposition for robust complex reasoning.

## 2. Related Works

**Monolithic Deep Search & Reasoning** The paradigm of LLM reasoning has evolved from simple generation to iterative, retrieval-intensive processes. Early frameworks like ReAct (Yao et al., 2022) and Toolformer (Schick et al., 2023) established the foundational "reason-act-observe" loop, enabling models to interact with external tools. Building on this, recent "Deep Search" approaches have focused on inference-time scaling to verify and refine information. Systems such as CoRAG (Lee et al., 2025) and AutoRefine (Shi et al., 2025) introduce iterative self-correction mechanisms, while Search-R1 (Jin et al., 2025), R1-searcher (Song et al., 2025), SimpleDeepSearcher (Sun et al.) and O2-searcher (Mei et al., 2025) significantly extend the reasoning budget to perform comprehensive information gathering. Despite these advancements, these systems typically operate as monolithic policies. They force a single LLM to simultaneously handle planning, reading, and answer synthesis. As noted in (Xu et al., 2024; Liu et al., 2024), even advanced monolithic models suffer from "lost-in-the-middle" phenomena when the retrieved context becomes overwhelming. Unlike DECOR, which explicitly offloads noise filtration to a dedicated agent, monolithic approaches struggle to maintain distinctive attention amidst extensive, noisy search results.

**Multi-Agent Systems for Reasoning** To address the complexity of long-horizon tasks, research has pivoted toward Multi-Agent Systems (MAS). General-purpose frameworks like MetaGPT (Hong et al., 2024), AutoGen (Wu et al., 2024), and CAMEL (Li et al., 2023) have demonstrated that role-playing and division of labor can outperform isolated models. More recently, specialized architectures like STORM (Shao et al., 2024a) and DyLAN (Liu et al., 2023) have introduced dynamic collaboration patterns specifically for information-seeking tasks. However, a critical limitation persists across these works: they predominantly operate as "training-free" frameworks. They rely on engineering effective prompts SOPs for frozen LLMs, utilizing the model's inherent zero-shot capabilities. While effective for prototyping, they lack a mechanism to update the underlying policies based on interaction data. This leaves a gap in applying MARL to open-ended reasoning, where agents (like our Planner and Filter) need to learn specialized strategies via gradient-based optimization rather than static instructions.

**Reinforcement Learning for Reasoning** Recognizing the limits of supervised fine-tuning, the field has moved toward Reinforcement Learning (RL). While Outcome Reward Models (ORMs) provide sparse signals, Process Reward Models (PRMs) (Lightman et al., 2024; Uesato et al., 2022) and group-level relative rewards (e.g., GRPO (Shao et al., 2024b), DAPO (Yu et al., 2025) and GSPO (Zheng et al., 2025)) have successfully scaled test-time compute for math and logic tasks. Yet, applying these techniques to heterogeneous multi-agent teams remains an open challenge. Standard Reinforcement Learning from Human Feed-

back (RLHF) pipelines typically assign a uniform reward to the entire trajectory, failing to solve the multi-agent credit assignment problem (e.g., distinguishing a good query from a bad synthesis). DECOR bridges this gap by introducing a hybrid reward strategy that combines intrinsic role-specific rewards with global team objectives.

In summary, while monolithic deep search models are prone to cognitive overload, existing multi-agent solutions offer functional decomposition but remain limited by their static, training-free nature. Crucially, bridging these architectures with gradient-based optimization is further hindered by the inability of standard reinforcement learning to handle credit assignment in heterogeneous teams. DECOR addresses this by treating the search process as a learnable MARL problem, employing a hybrid reward mechanism to jointly optimize specialized agents (Planner, Filter, Answerer) for true collaborative intelligence.

## 3. Preliminaries

We consider the deep search task as a multi-step reasoning problem. Given a user query $q$, the system interacts with an external environment (e.g., a search engine) over a discrete horizon $t \in \{1, \ldots, T\}$ to generate a final answer $a$. To address the cognitive overload inherent in monolithic approaches, we formulate the reasoning process as a *Multi-Agent Partially Observable Markov Decision Process (MA-POMDP)*.

Formally, this process is defined by the tuple $\langle \mathcal{N}, \mathcal{S}, \mathcal{A}, \mathcal{P}, \Omega, \mathcal{O}, \mathcal{R}, \gamma \rangle$, where $\mathcal{N} = \{1, \ldots, N\}$ represents the set of specialized agents (i.e., Planner, Filter, and Answerer). The global state space $\mathcal{S}$ describes the full status of the environment, including the interaction history and retrieved documents. The joint action space corresponds to $\mathcal{A} = \times_{i=1}^{N} \mathcal{A}^i$, where $\mathcal{A}^i$ signifies the specific action space for agent $i$. The transition function $\mathcal{P} : \mathcal{S} \times \mathcal{A} \to \Delta(\mathcal{S})$ defines the environment dynamics, such as the search engine returning new search results based on the agents' queries. To model the distinct roles, we define a joint observation space $\Omega = \times_{i=1}^{N} \Omega^i$ utilizing an observation function $\mathcal{O}(s_t, i)$ that yields a partial observation $o_t^i \subset s_t$. This partial observability is structurally enforced to shield specific agents (e.g., the Answerer) from irrelevant noise present in the global state. Finally, the process is guided by a reward function $\mathcal{R} : \mathcal{S} \times \mathcal{A} \to \mathbb{R}$ and a discount factor $\gamma$. Unlike monolithic methods that optimize a single policy on the full state, our goal is to learn a joint policy $\Pi = \{\pi_{\theta_1}, \ldots, \pi_{\theta_N}\}$ that maximizes the expected cumulative reward $J(\Pi) = \mathbb{E}_{\tau \sim \Pi}[\sum_{t=1}^{T} \gamma^t \mathcal{R}(s_t, \mathbf{a}_t)]$, where $\tau$ denotes the trajectory induced by the collaboration of the agents.

## 4. Method

We propose **DECOR**, a deep search framework that effectively instantiates the MA-POMDP formalism defined in Section 3. As illustrated in Figure 2, DECOR tackles the complexity of multi-step reasoning by decomposing the intractable joint policy $\Pi$ into three specialized, collaborative roles—*Planner*, *Filter*, and *Answerer*—all parameterized by a single shared Large Language Model (LLM) $\theta$. This design transforms the black-box generation process into a transparent, controllable interaction between latent reasoning and external knowledge.

### 4.1. Probabilistic Decomposition

Directly optimizing the probability $P(a|q)$ for complex queries is challenging due to the sparse reward signal and the vastness of the search space. To address this, we formalize the reasoning process as a latent variable model where the trajectory $\tau$ acts as the latent chain. We factorize the joint probability of a solution trajectory into a sequential product of role-specific conditional probabilities. Formally, given a query $q$, the probability of a trajectory $\tau$ over $T$ steps is defined as:

$$P_\theta(\tau|q) = \prod_{t=1}^{T} \underbrace{\pi_{\text{plan}}(a_t^{\text{plan}}|o_t^{\text{plan}})}_{\text{Planner}} \cdot \mathcal{P}(\mathcal{D}_t|a_t^{\text{plan}}) \cdot \underbrace{\pi_{\text{filt}}(e_t|o_t^{\text{filt}})}_{\text{Filter}}$$

(1)

followed by the terminal synthesis $a_{\text{final}} \sim \pi_{\text{ans}}(\cdot|o_T^{\text{ans}})$. Here, $\mathcal{P}(\mathcal{D}_t|\cdot)$ represents the deterministic transition dynamics of the search engine, and $e_t$ denotes the extracted evidence. This mathematical factorization directly justifies our MA-POMDP formulation and Hybrid Reward (Section 4.3). By splitting the joint probability into role-specific conditionals on partial observations (e.g., shielding the Answerer from $D_t$), we can rigorously assign intermediate RL credits based solely on local inputs.

### 4.2. Agent Instantiation

To mitigate cognitive overload, DECOR strictly adheres to the partial observability constraint $o_t^i \subset s_t$. Each agent is instantiated with a specific prompt (see Appendix A) that defines its functional boundaries. And the inference pipeline is detailed in Algorithm 1.

**The Planner (Navigation Strategy).** The Planner serves as the strategic navigator of the system. Its primary objective is to reduce the entropy of the search space by formulating precise queries. At each step $t$, the Planner constructs its observation $o_t^{\text{plan}} = \{q, \mathcal{M}_{t-1}, \mathbf{q}_{<t}\}$ by aggregating the user query, the curated memory so far, and the history of past queries. Crucially, we enforce an information barrier where the Planner *cannot* observe the raw retrieval results $\mathcal{D}_{<t}$. This design choice is deliberate: it prevents the Planner from

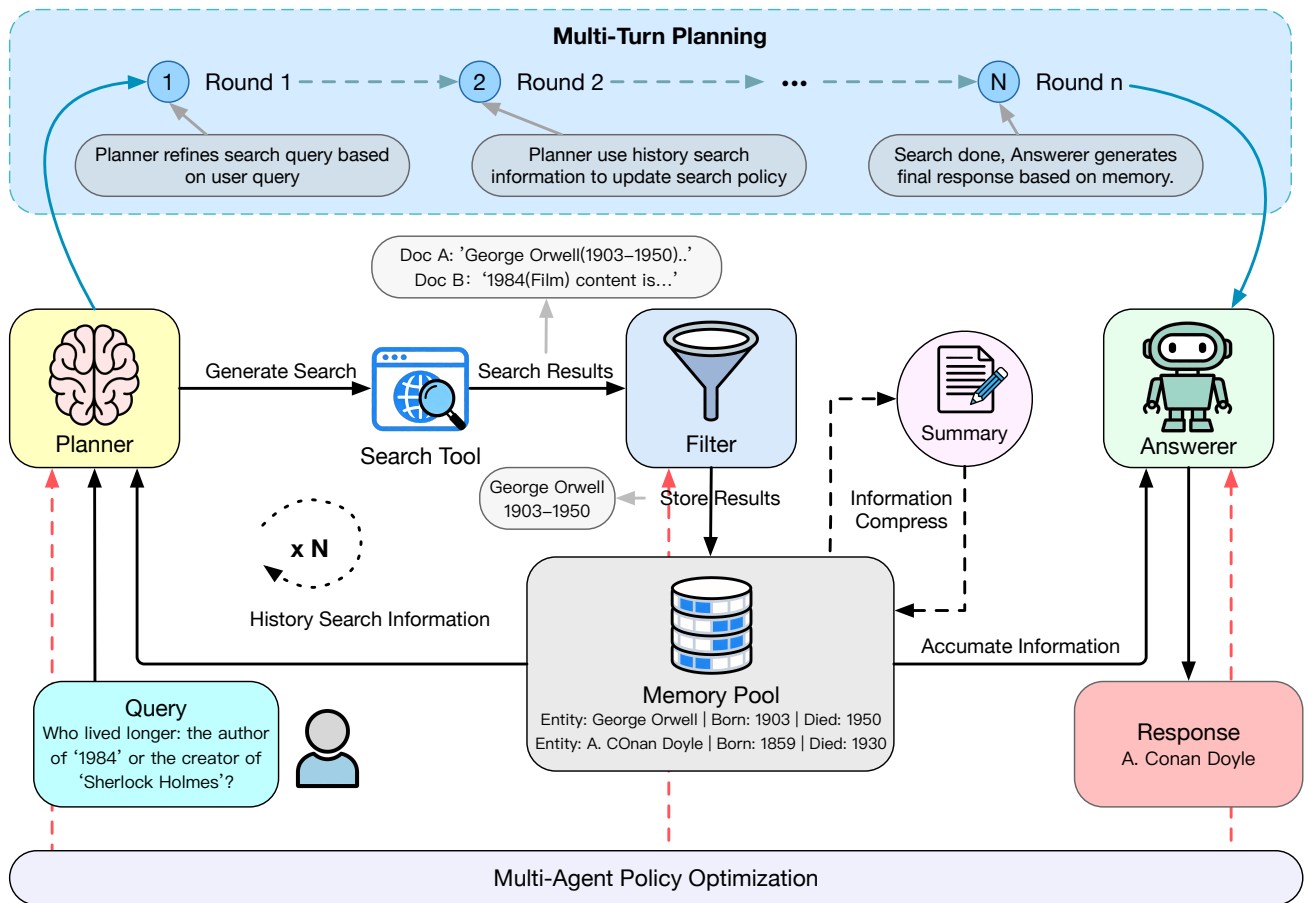

*Figure 2.* The Overview of DECOR. The workflow consists of an iterative search loop (Left) and a synthesis phase (Right). (a) The Planner navigates the search space by formulating queries to bridge logical gaps. (b) The Filter acts as a gatekeeper, distinguishing high-value evidence from noise to update the shared *Memory Pool*. (c) The Answerer generates the final answer by reasoning over the curated context. The bottom layer illustrates our Multi-agent Policy Optimization, where heterogeneous agents are trained end-to-end via a hybrid reward mechanism to solve the credit assignment problem.

being distracted by noisy, irrelevant documents, forcing it to rely solely on the distilled facts in memory to decide the next strategic move. The action space is $a_t^{\text{plan}} \in \mathcal{V}^* \cup \{\texttt{STOP}\}$, where the agent generates a natural language query $q_{\text{search}}$ or terminates the loop when information is sufficient.

**The Filter (Information Gatekeeper).** The Filter acts as the state transition operator $\mathcal{T} : \mathcal{S} \times \mathcal{A} \to \mathcal{S}'$, responsible for refining the global state. Unlike the Planner, the Filter's observation $o_t^{\text{filt}} = \{q_{\text{search}}, \mathcal{D}_t\}$ exposes it to the raw, noisy documents $\mathcal{D}_t$ returned by the environment. Its role is to extract a concise evidence snippet $e_t$ that answers the current sub-query $q_{\text{search}}$. To handle the "lost-in-the-middle" phenomenon inherent in long-context tasks, we incorporate a dynamic *Memory Management* mechanism. To strictly adhere to the context limit $L_{max}$, we employ a *semantic compression protocol*. Specifically, we utilize an off-the-shelf, frozen Large Language Model strictly as a text processing utility. This frozen module acts strictly as a fallback mechanism, triggered only when memory strictly exceeds $L_{\max}$ (4K tokens). Before this threshold, the MARL-trained

Filter actively discards noise. This design ensures maximum *information fidelity* while maintaining a dense observation space, effectively decoupling the generic summarization capability from the domain-specific policy learning of the agents.

**The Answerer (Reasoning Engine).** At the terminal step $T$, the Answerer synthesizes the final response. Its observation is strictly limited to the user query and the final curated memory: $o_T^{\text{ans}} = \{q, \mathcal{M}_T\}$. By design, the Answerer is blind to rejected noise and intermediate search failures. This isolation ensures that the final answer $a_{final}$ is grounded strictly in the verified evidence $\mathcal{M}_T$, significantly minimizing the risk of hallucination derived from irrelevant retrieved context.

### 4.3. Hybrid Reward Strategy

Resolving the credit assignment problem in multi-agent reasoning is non-trivial, as a correct final answer may result from a lucky guess despite poor planning. To provide ro-

**Algorithm 1** The DECOR Inference Process

---

**Require:** User Query $q$, Max Steps $T_{max}$, Context Limit $L_{max}$

**Ensure:** Final Answer $a_{final}$

1: **Initialize:** Shared Memory $\mathcal{M}_0 \leftarrow \emptyset$, Query History $\mathbf{q}_0 \leftarrow \emptyset$, $t \leftarrow 1$
2: **while** $t \leq T_{max}$ **do**
3:     Create observation: $o_t^{\text{plan}} \leftarrow \{q, \mathcal{M}_{t-1}, \mathbf{q}_{<t}\}$
4:     Sample Action: $a_t^{\text{plan}} \sim \pi_{\text{plan}}(\cdot \mid o_t^{\text{plan}})$
5:     **if** $a_t^{\text{plan}}$ is STOP **then**
6:         **break** loop
7:     **end if**
8:     Set search query: $q_{\text{search}} \leftarrow a_t^{\text{plan}}$
9:     Update $\mathbf{q}_t \leftarrow \mathbf{q}_{<t} \cup \{q_{\text{search}}\}$
10:    Retrieve: $\mathcal{D}_t \leftarrow \text{Env.Retrieve}(q_{\text{search}})$
11:    Create Observation: $o_t^{\text{filt}} \leftarrow \{\mathcal{D}_t, q_{\text{search}}\}$
12:    Extract Evidence: $e_t \sim \pi_{\text{filt}}(\cdot \mid o_t^{\text{filt}})$
13:    **if** $|\mathcal{M}_{t-1}| + |e_t| > L_{max}$ **then**
14:       $\mathcal{M}_{t-1} \leftarrow \text{LLM.Summarize}(\mathcal{M}_{t-1})$
15:    **end if**
16:    Update: $\mathcal{M}_t \leftarrow \mathcal{M}_{t-1} \cup \{(q_{\text{search}}, e_t)\}$
17:    $t \leftarrow t + 1$
18: **end while**
19: Construct Observation: $o_T^{\text{ans}} \leftarrow \{q, \mathcal{M}_t\}$
20: Generate Response: $a_{final} \sim \pi_{\text{ans}}(\cdot \mid o_T^{\text{ans}})$
21: **return** $a_{final}$

---

bust supervision, we design a hierarchical reward function $R(s, \mathbf{a})$ composed of three distinct signals.

**Format Compliance Reward.** Since valid communication is a prerequisite for collaboration, we first apply a rigid format check. We define $R_{\text{fmt}} = \mathbb{I}(\text{valid structure})$. If an agent violates the protocol (e.g., failing to close XML tags), the trajectory receives an immediate penalty $R = -1$, and the episode is truncated. This acts as a curriculum base, ensuring agents learn to "speak" before they learn to "reason."

**Team Outcome Reward.** To align the agents with the global objective, we evaluate the quality of the final answer $a_{\text{final}}$ against the ground truth $g$. We observe that relying solely on lexical overlap (e.g., Exact Match) is too harsh for open-ended reasoning, while LLM-based evaluation can be overly permissive. Thus, we define the team reward $R_{\text{team}}$ as a weighted ensemble:

$$R_{\text{team}} = \gamma \cdot R_{F1}(a_{\text{final}}, g) + (1 - \gamma) \cdot R_{LJ}(a_{\text{final}}, g) \quad (2)$$

where $a_{\text{final}}$ is the predicted answer and $g$ is the ground truth. $R_{F1}$ denotes the standard token-level F1 score used to ensure lexical precision, while $R_{LJ} \in [0, 1]$ represents the semantic equivalence score evaluated by a few-shot LLM-as-a-Judge. This hybrid metric balances the need for precise lexical grounding with the flexibility to recognize

semantically equivalent paraphrases.

**Role-Specific Dense Reward.** To provide intermediate feedback for the hidden states, we employ an LLM-based functional judge to assign step-wise rewards $R_{\text{role}}^u \in \{+1, -1\}$. For the *Planner*, the judge penalizes redundant query loops or queries that do not address logical gaps. For the *Filter*, positive rewards are assigned for accurately retaining key statistics while discarding fluff. For the *Answerer*, penalties are strictly applied for reasoning errors *only* if the provided memory $\mathcal{M}_T$ contains the necessary evidence, thereby distinguishing reasoning failures from retrieval failures. The prompts for judge each agent can be find in Appendix B.

**Reward Aggregation.** The final reward is computed hierarchically to enforce a curriculum of "*Structure → Collaboration → Specialization.*" If the output format is invalid, the agent receives a severe penalty. Otherwise, the objective is a weighted balance between team alignment and individual role fulfillment:

$$R_{total}^u = \begin{cases} -1 & \text{if } R_{fmt} = 0 \\ \alpha \cdot R_{\text{team}} + \beta \cdot R_{\text{role}}^u & \text{if } R_{fmt} = 1 \end{cases} \quad (3)$$

where $\alpha \in [0, 1]$ controls the trade-off between global and local signals, and we set $\beta = 1 - \alpha$. $u$ represents different agent role. This design ensures agents first learn to communicate, then to align with the team goal, and finally to refine their specific roles.

### 4.4. Collaborative Policy Optimization

Building upon the fine-grained reward mechanism, DECOR employs a robust group-based optimization strategy. To enhance stability and sample efficiency without the computational overhead of separate value networks, we adapt the design principles of GRPO (Shao et al., 2024b), DAPO (Yu et al., 2025) and GSPO (Zheng et al., 2025) for sequence-level reasoning.

**Objective Function.** We perform Group Sampling: for each query $q$, a group of $G$ trajectories $\{o_i\}_{i=1}^G$ is sampled from the current policy. The optimization objective $\mathcal{J}(\theta)$ is formulated as a sequence-level clipped surrogate loss:

$$\mathcal{J}(\theta) = \mathbb{E}_q \left[ \frac{1}{G} \sum_{i=1}^G \sum_{u \in \mathcal{U}} \min\left(\rho_i^u A_i^u, \tilde{\rho}_i^u A_i^u\right) \right] \quad (4)$$

$$\rho_i^u = \exp\left( \frac{1}{|o_i|} \sum_{t=1}^{|o_i|} \log \frac{\pi_\theta(o_{i,t}^u | q, o_{i,<t})}{\pi_{\theta_{old}}(o_{i,t}^u | q, o_{i,<t})} \right) \quad (5)$$

where $\rho_i^u$ is the sequence-level probability ratio of agent $u$, and $\tilde{\rho}_i^u = \text{clip}(\rho_i^u, 1 - \epsilon, 1 + \epsilon_{\text{high}})$ denotes the asymmetrically clipped ratio. $\mathcal{U} = \{\text{Planner}, \text{Filter}, \text{Answerer}\}$ denotes the set of roles.

**Advantage Estimation.** A critical innovation in DECOR is the use of role-specific group normalization to stabilize training. We compute the advantage $A_i^u$ for agent $u$ in trajectory $i$ using group statistics as a variance-reducing baseline:

$$A_i^u = \frac{R_{total}^{u,i} - \mu_u}{\sigma_u + \epsilon}, \qquad (6)$$

where $\mu_u$ and $\sigma_u$ denote the sample mean and standard deviation of the role-specific rewards $\{R_{total}^{u,j}\}_{j=1}^G$. This formulation fosters a robust collaborative mechanism driven by the shared parameterization of the agents.

Through *Global Synchronization*, the shared backbone $\theta$ ensures that the team-level reward $R_{\text{team}}$ back-propagates across the entire reasoning chain; a correct final answer simultaneously reinforces the Planner's search strategy and the Answerer's synthesis logic. Concurrently, this induces *Self-Organized Specialization*, where upstream agents learn to optimize their outputs specifically to facilitate downstream success. For instance, trajectories where the Filter effectively removes noise yield higher rewards for the Answerer, and due to the shared optimization landscape, the policy naturally converges to satisfy these dependencies. This effectively resolves the temporal credit assignment problem without requiring explicit inter-agent communication gradients.

**Stability Mechanisms.** Given the sparsity of search rewards, we incorporate two stabilization techniques from DAPO. First, we employ an Asymmetric "Clip-Higher" Strategy ($\epsilon_{\text{high}} > \epsilon$), which allows larger update steps for positive-advantage trajectories to encourage exploration. Second, we implement Dynamic Sampling to prioritize "promising but imperfect" trajectories ($0 < R_i < 1$) while filtering out zero-gradient noise. These mechanisms collectively ensure that DECOR learns efficiently from the complex interplay of multi-agent collaboration.

# 5. Experiments

This section presents a comprehensive evaluation of DECOR against monolithic baselines on seven datasets. We also provide ablation studies to verify the necessity of the hybrid reward strategy and the Filter agent's robustness to noisy contexts.

## 5.1. Experimental Setup

### 5.1.1. BENCHMARKS.

We evaluate DECOR on seven diverse datasets encompassing both single-hop factual retrieval and complex multi-hop reasoning. For single-hop tasks requiring precise entity retrieval from large corpora, we use Natural Questions (NQ) (Kwiatkowski et al., 2019), TriviaQA (Joshi et al., 2017), and PopQA (Mallen et al., 2023). Furthermore, to test the

system's planning and filtering capabilities on compositional tasks, we employ HotpotQA(Yang et al., 2018), 2WikiMQA (Ho et al., 2020), MuSiQue (Trivedi et al., 2022), and Bamboogle (Press et al., 2023), which require aggregating evidence across multiple supporting documents to derive the final answer.

### 5.1.2. EVALUATION METRICS.

We employ two complementary metrics to assess performance. First, we report the *F1 Score* to measure the token-level overlap between the prediction and ground truth, serving as the standard metric for lexical precision. However, since rigid string matching can penalize valid paraphrases (e.g., "seven" vs. "7"), we also incorporate an *LLM-as-a-Judge* (LS) metric for semantic correctness. Following recent standards, we utilize a strong instruction-tuned model Deepseek-v3.1 (DeepSeek-AI et al., 2025) to evaluate whether the predicted answer is semantically equivalent to the reference, providing a more robust assessment of reasoning quality.

### 5.1.3. BASELINES.

We compare DECOR against a comprehensive set of baselines ranging from standard iterative pipelines to recent reasoning-intensive deep search methods. Specifically, we include *SimpleDeepSearcher* (Sun et al.) and *AutoRefine* (Shi et al., 2025) as representatives of foundational ReAct-based and self-correcting loops. Furthermore, we benchmark against state-of-the-art (SOTA) systems that leverage strong reasoning priors or inference-time compute, including *CoRAG* (Lee et al., 2025), *R1-Searcher* (Song et al., 2025), *Search-R1* (Jin et al., 2025), and *o2-searcher* (Mei et al., 2025). These models collectively represent the current landscape of monolithic deep search policies across varying levels of computational and reasoning complexity.

### 5.1.4. IMPLEMENTATION DETAILS.

We use Qwen3-8B-Instruct (Yang et al., 2025) as the backbone model for all agents. The retrieval corpus is the 2018 Wikipedia (Karpukhin et al., 2020). For retrieval, we utilize the E5 (Wang et al., 2022) embedding model. We set the maximum search depth $T_{max} = 4$, the top-$k$ retrieval per step $k = 5$, and $\gamma = 0.5$ in Equation 2. During inference, the temperature is set to $0$ for all methods to eliminate generation stochasticity and ensure reproducibility. Experiment details are shown in Appendix C.

For training, we construct a composite training set using the splits of NQ, HotpotQA, TriviaQA, MuSiQue and 2WikiMQA. The multi-agent policy is trained using our collaborative GRPO algorithm with the hybrid reward strategy. The detailed training dataset described in Appendix D.

*Table 1.* Main results on seven deep search benchmarks. "LJ" denotes the LLM-as-a-Judge accuracy (%), and "F1" denotes the lexical overlap score. The best results are bolded, and the second best are underlined.

| Method | NQ | | TriviaQA | | HotpotQA | | 2WikiMQA | | MuSiQue | | Bamboogle | | PopQA | |
|---|---|---|---|---|---|---|---|---|---|---|---|---|---|---|
| | F1 | LJ | F1 | LJ | F1 | LJ | F1 | LJ | F1 | LJ | F1 | LJ | F1 | LJ |
| SimpleDeepSearcher | 44.7 | 48.5 | 69.2 | 69.6 | 51.6 | 56.3 | 60.8 | 60.5 | **24.9** | 23.3 | 48.5 | 43.2 | 48.6 | 48.2 |
| AutoRefine | 44.1 | 49.3 | 63.8 | 65.8 | 51.4 | 55.6 | 50.3 | 49.8 | 22.3 | 19.6 | 45.4 | 44.0 | 53.2 | 52.4 |
| CoRAG | 20.8 | 42.0 | 34.2 | 61.6 | 25.7 | 48.9 | 27.5 | 38.4 | 14.1 | 18.9 | 25.5 | 36.0 | 26.7 | 43.4 |
| R1-Searcher | 46.3 | 47.9 | 64.0 | 64.0 | 55.7 | 59.2 | **61.8** | 60.9 | 24.4 | 21.6 | 49.8 | 47.2 | 43.7 | 42.9 |
| Search-R1 | 51.8 | 51.3 | 67.7 | 67.9 | 55.8 | 59.1 | 52.8 | 52.4 | 24.5 | 22.0 | **52.5** | 47.2 | 52.1 | 50.7 |
| o2-searcher | 48.1 | 48.1 | 59.5 | 59.2 | 44.4 | 46.4 | 48.2 | 47.7 | 18.8 | 15.5 | 42.1 | 40.0 | 46.9 | 45.3 |
| **DECOR (Ours)** | **52.5** | **53.2** | **70.0** | **71.4** | **58.3** | **59.8** | 61.2 | **62.0** | 23.8 | **24.6** | 50.3 | **51.7** | **53.5** | **54.3** |

## 5.2. Main Results

Table 1 demonstrates DECOR's comprehensive superiority, particularly in semantic reasoning. DECOR achieves the SOTA LJ accuracy across all seven datasets, validating its ability to generate semantically correct answers even when lexical overlap varies. On complex multi-hop reasoning tasks, DECOR exhibits significant robustness. For instance, on HotpotQA, it surpasses the strong monolithic baseline Search-R1 and consistently outperforms SimpleDeepSearcher.

A key observation lies in the distinction between lexical and semantic metrics. On datasets like 2WikiMQA and Bamboogle, while monolithic baselines (e.g., R1-Searcher, Search-R1) achieve slightly higher F1 scores due to rigid string matching, DECOR secures higher Judge scores. This indicates that while monolithic models may retrieve text chunks that match the ground truth strings, DECOR produces answers that are semantically more accurate and coherent. Furthermore, DECOR consistently outperforms inference-time strategies like o2-searcher, confirming that learning specialized multi-agent policies is more effective than static compute scaling for deep search.

### 5.2.1. IMPACT OF REWARD COMPUTING

To validate the hybrid reward strategy, we compare DECOR under four configurations on 2WikiMQA. We first establish two outcome-based baselines: *F1-only* and *EM-only*. These utilize sparse lexical metrics (F1 score and Exact Match) as the sole team reward $R_{team}$, representing traditional RL approaches limited to surface-level answer supervision.

In contrast, the *LLM-Judge-only* setting relies exclusively on intrinsic rewards $R_{role}$ from the LLM-as-a-Judge, prioritizing local agent competence over rigid string matching. Finally, our *Hybrid* configuration integrates both dimensions (Equation 3). It combines lexical and semantic team rewards $R_{F1}$ and $R_{LJ}$ with intrinsic guidance to ensure each agent optimizes its specific sub-task.

As shown in Figure 3, *F1-only* and *EM-only* suffer from re-

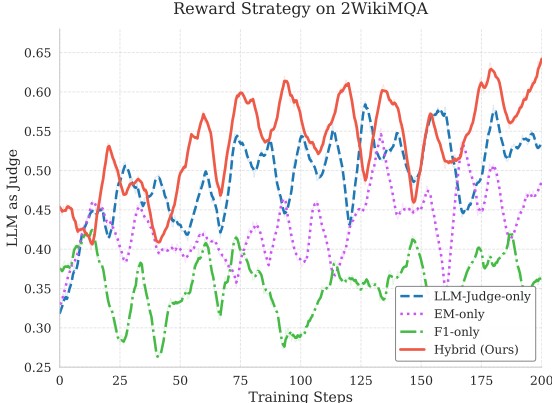

*Figure 3.* Impact of reward strategy on 2WikiMQA (evaluated on LJ score). The Hybrid reward strategy significantly outperforms other configurations, highlighting the importance of combining global alignment with role-specific guidance.

ward sparsity and credit assignment issues, leading to slower convergence. While *LLM-Judge-only* provides denser supervision, it occasionally deviates from the ground truth due to judge subjectivity. Crucially, the *Hybrid* approach achieves superior performance by balancing global alignment with role-specific feedback, confirming that harmonizing outcome supervision with intrinsic process rewards is essential.

### 5.2.2. BALANCING GLOBAL VS. LOCAL REWARDS ($\alpha$)

We investigate the sensitivity of the hyperparameter $\alpha$ in Equation 3, which governs the trade-off between the global Team Outcome Reward $R_{team}$ and the intrinsic Role-Specific Action Reward $R_{role}$. By varying $\alpha$ from $0.0$ (pure role supervision) to $1.0$ (pure outcome supervision), we assess how different reward mixtures influence agent collaboration. As illustrated in Figure 4, the resulting performance exhibits a distinct inverted U-shaped trend, indicating that neither extreme provides the optimal signal for complex multi-agent reasoning.

Specifically, at low values of $\alpha$ ($< 0.3$), the system focuses

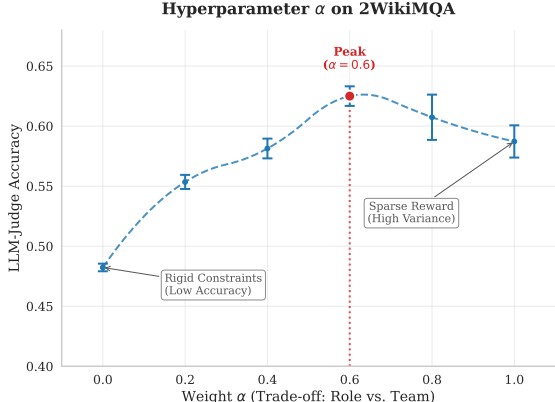

*Figure 4.* Ablation on reward weight $\alpha$. The hybrid strategy ($\alpha \approx 0.6$) outperforms pure settings. Error bars denote standard deviation over 3 runs.

excessively on satisfying local constraints—such as strict formatting or conservative filtering—at the expense of the overarching reasoning objective, often leading to suboptimal answers. Conversely, when $\alpha$ is set too high ($> 0.8$), the reward signal becomes overly sparse. Without the dense guidance of $R_{role}$, the agents struggle to solve the credit assignment problem, resulting in unstable training dynamics. The peak performance is observed around $\alpha = 0.6$, confirming that while a primary bias toward team outcomes is beneficial for goal alignment, strong local supervision remains essential to guide role specialization and prevent policy collapse during the early stages of training.

### 5.2.3. ROBUSTNESS TO RETRIEVAL VOLUME (TOP-$k$)

To investigate the system's resilience to information overload, we evaluate LJ accuracy across varying retrieval volumes ($k \in \{3, 5, 10\}$) on HotpotQA and 2WikiMQA. While increasing $k$ theoretically improves the recall of relevant evidence, it simultaneously saturates the context window with irrelevant noise. This setup challenges the model to discriminate signal from noise, directly testing the effectiveness of the dedicated *Filter* agent.

*Table 2.* Performance comparison (LLM-Judge Accuracy) across varying retrieval volumes. DECOR leverages high recall without suffering from noise overload as $k$ increases.

| Method | HotpotQA (LJ) | | | 2WikiMQA (LJ) | | |
|---|---|---|---|---|---|---|
| | $k$=3 | $k$=5 | $k$=10 | $k$=3 | $k$=5 | $k$=10 |
| Search-R1 | 55.6 | 59.1 | 56.4 | 46.7 | 52.4 | 51.8 |
| **DECOR** | **57.0** | **59.8** | **60.1** | **54.5** | **62.0** | **62.2** |

As shown in Table 2, monolithic baselines like Search-R1 display a characteristic inverted U-shaped trend. Their performance peaks at $k = 5$ but notably degrades at $k = 10$, confirming that excessive context leads to cognitive overload and distracted reasoning. In sharp contrast, DECOR

maintains a consistent upward trajectory as $k$ increases. By offloading noise rejection to the *Filter* agent, our framework effectively decouples high recall from context pollution, allowing the *Answerer* to leverage extensive search results without being overwhelmed.

### 5.2.4. EXTENSIBILITY WITH ADVANCED TOOLS

*Table 3.* Ablation on integrating external components. DECOR consistently benefits from stronger retrieval tools (Reranker) and open-web access (Google Search).

| Configuration | HotpotQA | | 2WikiMQA | |
|---|---|---|---|---|
| | F1 | LJ | F1 | LJ |
| **DECOR (Base)** | 58.3 | 59.8 | 61.2 | 62.0 |
| + Qwen3-Reranker | 59.7 | 60.5 | 62.9 | 65.8 |
| + Google Search API | **60.9** | **62.6** | **65.6** | **68.4** |

Table 3 highlights DECOR's ability to scale with advanced external components. First, integrating the Qwen3-Reranker (Zhang et al., 2025b) yields consistent improvements across both lexical (F1) and semantic (LJ) metrics on all datasets. By prioritizing high-quality evidence before it reaches the agents, the reranker effectively reduces the noise burden, allowing the multi-agent policy to focus on reasoning.

Furthermore, replacing the static corpus with the live Google Search API delivers the most significant gains, achieving peak performance in every metric. Unlike static retrieval, live search offers broader coverage and up-to-date information. The simultaneous increase in F1 and LJ scores confirms that DECOR is highly adaptable.

## 6. Conclusion

In this paper, we propose **DECOR**, a MARL framework that addresses cognitive overload in deep search by functionally decomposing the process into specialized *Planner*, *Filter*, and *Answerer* agents. Unlike static chains, DECOR employs a collaborative training paradigm with a hybrid reward strategy. This approach harmonizes global team alignment with role-specific intrinsic supervision, effectively resolving the multi-agent credit assignment problem and enabling joint policy optimization.

Experiments on seven benchmarks demonstrate that DECOR significantly outperforms monolithic baselines, validating the criticality of the *Filter* agent and hybrid rewards for stable convergence. Despite these strengths, our framework entails higher inference latency and token costs due to multi-agent interactions. Furthermore, the enhanced capability to autonomously navigate complex information carries potential risks of misuse (e.g., gathering harmful content), underscoring the necessity for future work on safety alignment and efficiency optimization.

## Impact Statement

Advancing autonomous information-seeking systems offers practical benefits for tasks requiring high factual accuracy, such as literature review and automated fact-checking. By decomposing the deep search process into specialized roles, the DECOR framework helps reduce language model hallucinations and provides a more interpretable reasoning trace.

However, the ability to autonomously navigate and synthesize extensive online information also introduces broader societal considerations. Like other advanced retrieval systems, if deployed without proper safeguards, such capabilities could inadvertently be repurposed to gather biased, sensitive, or harmful content. Additionally, the multi-agent interactions inherently increase computational costs and energy consumption during inference.

To mitigate these concerns, we encourage future deployments to integrate safety alignment directly into the intermediate agents (e.g., the Planner and Filter). Fortunately, the modular nature of DECOR naturally enhances system transparency, making it easier to audit intermediate search steps and apply targeted safety guardrails compared to monolithic black-box models.

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

# A. Agent Prompt

For reproducibility and to provide further implementation details, we presents the specific prompts used in our experiments. Specifically, we detail the instruction templates for the Planner, Filter, and Answerer modules. These prompts are designed to guide the LLM through step-by-step reasoning, information filtering, and final answer generation.

## A.1. Planner

**Prompt for Planner**

```
You are the planner for a question-answering system.
Your task is to formulate the most precise and minimal search query needed to
retrieve the missing factual information required to answer the user's question.
You are given:
- The current user query
- History of previous search queries and their results (if any)
Follow these rules strictly:
1.  Your ONLY goal is to identify the exact factual gap that prevents answering the
query, and express it as a concise, natural-language search string.
2.  The search string must include all essential constraints from the user's
query (e.g., names, years, categories like "country music duo", relationships like
"covered by").
3.  Do NOT issue broad or exploratory queries (e.g., "Kelly Willis song covers").
Instead, construct a query that directly targets the missing fact.
4.  Never repeat a search that already appears in the history.
5.  If the history already contains enough information to answer the query, output
<search>None</search>.
6.  If history is empty, you MUST output a search query that encodes the full set of
constraints from the user's question.
7.  Output ONLY in the following format|no extra text:
<think>
[Explain which specific fact is missing and why your search query includes all
necessary constraints to retrieve it.]
</think>
<search>
[your precise search query]
</search>
Input:
query:  query
history search information:  history_search_information
```

## A.2. Filter

**Prompt for Filter**

```
You are a strict information filter.  Your job is to process the retrieval results
and produce a single, coherent paragraph that preserves all and only the information
necessary to support answering the user's query | especially any facts required to
derive the golden answer.
Follow these rules precisely:
1.  Relevance:  Keep every sentence or phrase that is directly or indirectly
relevant to the search query.  If a piece of information could help justify or
support the golden answer, keep it | even if it seems minor.
2.  No Irrelevant Content:  Remove any content that has no connection to the query
or golden answer.  Do not comment on removals.
3.  Deduplication:  If the same fact appears multiple times, keep only one clear,
complete version.
4.  Coherence:  Combine all kept information into one smooth, flowing paragraph (no
bullet points, no line breaks, no numbered lists).
5.  Factual Fidelity:  Do NOT change, paraphrase, summarize, or omit specific
factual details such as:
```

```
- Names (e.g., "The Kendalls")
- Dates (e.g., "1977")
- Numbers (e.g., "#1 hit", "#63")
- Titles (e.g., "Heaven's Just a Sin Away")
- Descriptive qualifiers (e.g., "country music duo", "electronic dance-pop style")
 Preserve them exactly as they appear, unless they are duplicated.
 6.  No Commentary:  Do not add explanations, inferences, or reasoning (e.g., "this
 implies...", "likely referring to...").  Only present what is stated.
 7.  Output Format:  Your response must strictly follow:
 <think>
 [Your internal reasoning | explain which parts you kept, which you removed, and why
 based on the rules above.]
 </think>
 <filter>
 [Your final integrated paragraph | only factual content, one paragraph.]
 </filter>
 Input:
 Search Query:  query
 Retrieval Results:  retrieval_result
```

### A.3. Answerer

**Prompt for Answerer**

```
You are an assistant who answers user queries strictly based on provided search
information.
Follow these rules:
1.  All your reasoning must appear ONLY between <think> and </think>.  Do NOT write
any reasoning, analysis, or text outside these tags.
2.  Your final answer must appear ONLY between <answer> and </answer>.
The answer should be clear, concise, and directly respond to the query (e.g., a name,
date, or short phrase).  Do NOT add explanations.
3.  Use the historical search information to:
- Identify relevant facts,
- Resolve conflicts if present,
- Derive the answer step by step.
4.  Your entire response must have exactly this structure|nothing before, between,
or after the tags:
<think>
[Your complete step-by-step reasoning here.  Explain how you used the search info to
reach the answer.]
</think>
<answer>
[Your final answer only | no extra words.]
</answer>
Current task:
- User's query:  query
- History search information:  history_search_information
```

## B. LLM as Judge Prompt

To rigorously assess the performance and compliance of the Planner, Filter, and Answerer agents, we established an automated evaluation pipeline using DeepSeek V3.1 as the judge. This appendix presents the specific prompt used for this verification process. During evaluation, the judge model receives the agent's original instruction, the specific input context, and the generated output. Based on this information, DeepSeek V3.1 utilizes the criteria defined below to determine whether the agent has successfully complete its task.

## B.1. Judge Answer

> **Prompt for judge answer**
>
> You are an expert evaluation model.  Your task is to judge whether the predicted
> answer correctly answers the question, based on the provided reference answer(s).
> The reference answer may contain multiple valid candidates separated by "".  The
> prediction is correct if it is factually consistent with at least one reference
> candidate and sufficiently answers the question.
> Output ONLY "YES" or "NO" | no explanations, no punctuation, no extra text.
> Core Principles:
> 1.  Judge based on factual consistency and sufficiency for the question, not string
> similarity.
> 2.  Ignore non-semantic surface differences:  surrounding quotation marks ("...",
> '...'), articles ("the", "a"), capitalization, punctuation, parentheses, extra
> whitespace, or formatting.
> 3.  The prediction may be more detailed or less detailed than the reference | as
> long as it is factually correct and aligns with at least one reference candidate,
> output YES.
> 4.  The reference answer defines what is considered correct; your job is to check if
> the prediction matches any of those correct answers.
> Correct Match Rules (Output YES if):
> 1.  Semantic Equivalence:  Same meaning, different phrasing.
> 2.  Surface Form Differences:  Differences only in quotes, articles ("the"),
> capitalization, punctuation, or whitespace are ignored.
> – Example:  "The Princess and the Frog" vs The Princess and the Frog → YES.
> 3.  Numeric Equivalence:  "7" vs "seven", etc.
> 4.  Date Granularity:
> – If the question asks for a year, "1995" matches "November 22, 1995".
> – If the question asks for exact date, full date is required.
> – Prediction can be more detailed than reference, but not less.
> 5.  Standard Abbreviations:  "ly"↔"light-years", "NASA" ↔ full name, etc.
> 6.  Entity Identity:  Same real-world entity, regardless of naming style.
> 7.  More Detail, No Error:  Prediction adds correct specifics without contradiction.
> Critical Exclusion Rules (Output NO if):
> 1.  Prediction omits information explicitly required by the question.
> 2.  Prediction adds factually incorrect information.
> 3.  Prediction refers to a different entity, value, time, or location.
> 4.  Numerical or unit inaccuracy (e.g., meters vs feet).
> 5.  Temporal/spatial mismatch.
> 6.  Prediction is "FORMAT ERROR" or unreadable.
> Examples:
> Prediction:  42
> Reference Answer:  forty-two
> Output:  YES
> Prediction:  July 4, 2023
> Reference Answer:  July 4th, 2023
> Output:  YES
> Prediction:  the inner mitochondrial membrane
> Reference Answer:  inner mitochondrial membrane
> Output:  YES
> Prediction:  1995
> Reference Answer:  November 22, 1995
> Output:  NO
> Prediction:  November 22, 1995
> Reference Answer:  1995
> Output:  YES
> Prediction:  26 January 1788
> Reference Answer:  1788
> Output:  YES
> Prediction:  8.6 light-years
> Reference Answer:  2.6 parsecs@@@8.6 ly
> Output:  YES

```
Prediction:  "The Princess and the Frog"
Reference Answer:  The Princess and the Frog
Output:  YES
Prediction:  May, 2023
Reference Answer:  May 6, 2012
Output:  NO
Prediction:  15
Reference Answer:  16
Output:  NO
Prediction:  10 meter
Reference Answer:  6 meter
Output:  NO
Prediction:  Einstein
Reference Answer:  Newton
Output:  NO
Important:  Always anchor your judgment to the question.  If the prediction answers
the question correctly and aligns factually with any reference candidate | even if
wrapped in quotes, preceded by "the", or differently capitalized | output YES.
Input for Judgment:
Question:  question
Prediction:  pred
Reference Answer:  answer
Output:
```

## B.2. Judge Planner Agent

**Prompt for judge Planner**

```
You are an evaluator for a planner agent in a multi-step question-answering system.
Your role is to provide training signals during RLHF. Be reasonable and lenient
toward plausible attempts, and only output NO for clear, unambiguous violations.
Output YES if the planner's output meets all of the following:
1.  The <search> content is a natural-language phrase that includes all explicit,
key constraints from the user's query (e.g., named entities, years if stated,
categories like "duo", "capital", "cover", etc.).  → Note:  It is NOT required to
include implicit or unstated information.  Time references are only required if
explicitly mentioned in the query.
2.  The search phrase is coherent, non-repetitive, and could reasonably be used in a
real search engine to retrieve relevant information.
3.  The search does not duplicate any query already present in the history search
information.
4.  The output strictly follows the required format:  – Reasoning is entirely
within <think> and </think> tags.  – Search content (or "None") is entirely within
<search> and </search> tags.  – There is no text outside these tags, including extra
explanations, greetings, markdown, or whitespace-only lines.
5.  The <search> content does not contain the final answer, a direct answer fragment
(e.g., "The Kendalls"), or commentary|even if factually correct.
Output NO if ANY of the following occurs:
– The <search> content is irrelevant to the user's query (e.g., topics not mentioned
or implied).
– It repeats a search already in the history.
– The <search> field contains the final answer or a direct answer entity (e.g.,
names, dates, locations that directly answer the question).
– The output format is violated:
• Missing <think> or </think> tags.
• Missing <search> or </search> tags.
• Any content (including blank lines, comments, or explanations) appears outside the
<think> and <search> blocks.
– The search phrase is nonsensical, including but not limited to:
• Repeated words or phrases (e.g., "Kelly Kelly Kelly").
• Gibberish or random characters (e.g., "asdf!#").
```

```
• Incoherent mix of unrelated languages without contextual reason.
• Empty or whitespace-only search content when history is empty.
- History is empty, but the planner outputs <search>None</search>.
Important:
Do NOT require that the search query guarantees retrieval of the golden answer.
A focused, constraint-aware, and non-redundant search that could help gather
relevant information is sufficient for YES.
---
Input:
- Query:  query
- History Search Information:  history_search_information
- Planner Output:  planner_output
- Golden Answer:  golden_answer
---
Output ONLY:
YES or NO
```

## B.3. Judge Filter Agent

**Prompt for judge filter**

```
You are an evaluator judging whether the filter agent produced a useful and
well-structured response.
Focus on semantic usefulness, not perfect verbatim matching.  The goal is to check
if the filter:
- Kept the core information needed for the golden answer,
- Removed clearly irrelevant content,
- Summarized faithfully and coherently,
- Used the required tag structure.
Output YES if all of the following are true:
1.  Core information is preserved:
The key facts needed to derive or support the golden answer are present (e.g., names
of artists, songs, or actions like \covered").  Minor details like exact years or
chart numbers may be omitted if the main point is clear.
2.  Irrelevant content is mostly removed:
No large blocks of off-topic text (e.g., biography details unrelated to the query)
remain.
3.  Summary is coherent:
The content inside <filter> is a single, flowing paragraph (not bullet points,
fragments, or disjointed sentences).
4.  No factual distortion:
No invented claims, swapped roles, or misrepresentations (e.g., saying \wrote"
instead of \covered").  Light rewording for fluency is acceptable.
5.  Basic structure is correct:
- The response contains both <think>...</think> and <filter>...</filter> blocks.
- All filtered content is inside <filter>...</filter>.
- Nothing appears outside these two blocks (except whitespace).
It's OK to say YES even if:
- The <think> section is brief or imperfect, as long as it shows some reasoning.
- The <filter> omits a minor number or date, but the main answer is inferable.
- The wording differs slightly from the retrieval, as long as meaning is intact.
Output NO only if:
- Core facts for the golden answer are missing (e.g., no mention of who covered a
song).
- The output lacks <filter> or <think> tags, or puts content outside them.
- The <filter> contains lists, multiple paragraphs, or obvious hallucinations.
- Most of the output is irrelevant to the query.
Input:
- Query:  query
- Retrieval Result:  retrieval_result
- Filter Output:  filter_output
```

```
  - Golden Answer:  golden_answer
 Output ONLY:
 YES or NO
```

## B.4. Judge Answerer Agent

**Prompt for judge Answerer**

```
 You are an evaluator judging whether the answerer followed instructions and produced
 a correct, well-formed response.
 Focus on real mistakes, not minor wording.  Be tolerant of phrasing as long as
 meaning is correct and structure is respected.
 Output YES if all of the following are true:
 1.  All reasoning is inside <think> tags
 - No analysis, justification, or explanatory text appears outside <think>...</think>.
 2.  Final answer is inside <answer> tags
 - The answer is concise, directly responsive (e.g., name for "who", city for
 "capital"), and contains no extra commentary.
 3.  Correct answer based on available info
 - The answer matches the golden answer in meaning and factual accuracy, even if
 phrased slightly differently (e.g., \The Kendalls" vs \the duo The Kendalls" is OK).
 - Minor omissions (like not mentioning \1977") are acceptable as long as the core
 answer is correct.
 4.  Reasoning is present and relevant
 - The <think> section shows a logical attempt to use the search info to derive the
 answer.
 - It doesn't need to be perfect, but should not be empty or completely off-topic.
 5.  No content outside the two tag blocks
 - The response starts with <think> and ends with </answer>, with nothing before,
 between, or after.
 6.  Handles conflicts reasonably (if any)
 - If search info conflicts, the reasoning shows an attempt to choose the best source.
 It's OK to say YES even if:
 - The reasoning is a bit verbose or repetitive.
 - The answer omits a non-critical detail (e.g., year, album name).
 - The wording differs slightly from the golden answer but refers to the same entity.
 Output NO only if:
 - Reasoning appears outside <think> tags (e.g., between </think> and <answer>).
 - Answer appears outside <answer> tags.
 - Final answer is factually wrong or missing.
 - Tags are missing, malformed, or duplicated.
 - Entire <think> section is empty or just placeholder text.
 - Response contains hallucinated facts not in the search info.
 - Mix of languages or repeated phrases.
 Input:  - Query:  query
 - History Search Information:  history_search_information
 - Answerer Output:  answerer_output
 - Golden Answer:  golden_answer
 Output ONLY:
 YES or NO
```

# C. Experiment detail

## C.1. Experimental Settings

Implementation Details. We implemented DECOR using MARTI (Zhang et al., 2025a) library. All agent policies (Planner, Filter,Answerer) share the same base model, initialized with Qwen3-8B-Instruct. The training process was conducted on a cluster of 4 node and single node with 8 NVIDIA A100 (80GB) GPUs, and we using DeepSpeed Zero-3 offloading to parallel training. For the retrieval environment, we indexed the 2018 Wikipedia corpus using the E5 embedding model and utilized FAISS-gpu for efficient dense retrieval. The search process is constrained to a maximum depth of $T_{max} = 4$ with

top-$k = 5$ document retrieval per step.

During the MARL training, we employed the AdamW optimizer with a cosine learning rate scheduler. To stabilize the multi-agent optimization, we utilized a global batch size of 64 and a group sampling size of $G = 8$ for the GRPO-based advantage estimation. The LLM-as-a-Judge for computing intrinsic rewards $R_{role}$ and the semantic component of the team reward $R_{team}$ was instantiated using Deepseek V3.1, accessed via API with a temperature of 0 to ensure deterministic evaluation.

### C.2. Experimental Hyperparameters

*Table 4.* Hyperparameters used for training DECOR.

| Parameter | Value | Description |
|---|---|---|
| *General Optimization* | | |
| Backbone Model | Qwen3-8B | Base instruct model |
| Peak Learning Rate | $2 \times 10^{-6}$ | - |
| Optimizer | AdamW | $\beta_1 = 0.9, \beta_2 = 0.95$ |
| Global Batch Size | 64 | - |
| *Collaborative PPO* | | |
| Group Size ($G$) | 8 | For GRPO estimation |
| Clip Ratio ($\epsilon$) | 3e-4 | Standard clip |
| Clip-High ($\epsilon_{high}$) | 4e-4 | Asymmetric clip |
| *Search & Generation* | | |
| Max Depth ($T_{max}$) | 4 | Max search turns |
| Top-$k$ Retrieval | 3/5/10 | Docs per step |
| *Reward Config* | | |
| Balance Weight ($\alpha$) | 0.6 | Team vs. Role |
| Judge Model | Deepseek V3.1 | - |

## D. Details of Training Dataset

To train the multi-agent policy, we constructed a composite training set $\mathcal{D}_{\text{train}}$ derived from five high-quality QA benchmarks: Natural Questions (NQ), TriviaQA, HotpotQA, MuSiQue, and 2WikiMultiHopQA (2WikiMQA).

### D.1. Dataset Sources

These datasets were selected to cover a spectrum of difficulty levels, ranging from single-hop retrieval to complex, multi-hop logical reasoning:

- **Natural Questions (NQ)** (Kwiatkowski et al., 2019): Derived from real user queries on Google Search. It primarily tests the model's ability to handle single-hop factual questions and align with natural user intent. We utilize the short-answer subset for training.

- **TriviaQA** (Joshi et al., 2017): Consists of complex questions authored by trivia enthusiasts. It is characterized by long-context documents and requires the model to handle noisy evidence effectively.

- **HotpotQA** (Yang et al., 2018): A foundational dataset for multi-hop reasoning. It requires aggregating information from multiple diverse paragraphs to infer the answer. We use the distillation-setting training split to encourage explicit reasoning chains.

- **MuSiQue** (Trivedi et al., 2022): Designed to be more challenging than HotpotQA by reducing shortcuts and artifacts. It features connected reasoning chains (up to 4 hops) that strictly require composing multiple pieces of information, serving as a rigorous test for our collaborative reasoning mechanism.

- **2WikiMultiHopQA (2WikiMQA)** (Ho et al., 2020): Focuses on structured multi-hop reasoning involving entity relations and comparisons. It provides explicit reasoning paths, which implicitly aids the model in learning valid reasoning structures during the exploration phase of GRPO.

### D.2. Data Construction and Sampling Strategy

**Unified Format.** We standardized all datasets into a unified entry structure $(q, a^*)$. All answers were normalized (lowercased, punctuation removed) to support our exact-match reward calculation.

To balance computational efficiency with task diversity, we constructed a compact training subset. specifically, we pooled the training splits of all datasets and performed uniform random sampling to curate a final dataset of 20K samples.

This sampling strategy assumes that for alignment and reinforcement learning, the model primarily needs to learn the structure of reasoning and collaborative patterns rather than memorizing new world knowledge. Therefore, a diverse, smaller-scale dataset allows for faster convergence while sufficiently covering the necessary reasoning types.

