# OpenReview forum: "DECOR: Learning to Decompose and Collaborate in Deep Search via Multi-Agent Reinforcement Learning"
_ICML.cc/2026/Conference — ICML 2026 regular_

### Official Review · Reviewer_xMY4 · 2026-03-07

**Soundness:** 2
**Presentation:** 2
**Significance:** 2
**Originality:** 2
**Overall Recommendation:** 3
**Confidence:** 3

**Summary:**

DECOR is a deep search framework based on MARL. It decomposes complex reasoning tasks into three specialized roles (Planner, Filter, and Answerer) and uses a hybrid reward strategy that combines role‑specific intrinsic feedback with team‑level outcome signals for joint optimization. By doing so, it effectively mitigates the “cognitive overload” and hallucination issues that single agents often encounter when handling large volumes of search results. DECOR outperforms existing single‑agent models and static multi‑agent systems across seven mainstream benchmark tests.

**Compliance With Llm Reviewing Policy:**

Affirmed.

**Ethical Review Concerns:**

DECOR greatly enhances a system’s ability to autonomously navigate and integrate complex information. The authors note that such powerful retrieval and synthesis capabilities could be misused. For example, to automatically collect harmful content or perform targeted extraction of sensitive information. Therefore, the framework still requires further constraints and research in safety alignment.

**Ethical Review Flag:**

Flag this paper for an ethics review.

**Ethics Expertise Needed:**

["Inappropriate Potential Applications & Impact (e.g., human rights concerns)"]

**Final Justification:**

Although role-specific rewards are introduced, the root causes of failures in complex long-horizon tasks remain highly coupled across agents. The authors’ response does not convincingly demonstrate that this mechanism fundamentally resolves issues such as inter-agent blame shifting or the accumulation and propagation of errors.

Although I have adjusted my score upwards, I still maintain a negative recommendation for this paper.

**Key Questions For Authors:**

DECOR decomposes the task into three roles: Planner, Filter, and Answerer. How exactly does this role‑based division of labor, especially the introduction of the Filter agent, alleviate the “cognitive overload” mentioned in the paper and address the common “lost‑in‑the‑middle” phenomenon that single models experience when dealing with long‑context retrieval?

How does the Hybrid Reward strategy balance global and local objectives? The paper introduces a reward mechanism that combines Team Outcome–level signals with Role‑Specific intrinsic rewards. How does this design address the classic “credit assignment” problem in multi‑agent collaboration? Furthermore, what does the observed “inverted‑U” trend of the hyperparameter $\alpha$, which balances these two reward components, imply?

What is unique about DECOR’s performance under large‑scale retrieval noise? In robustness experiments with varying retrieval sizes (Top‑k), why do single‑agent baselines (such as Search‑R1) show performance degradation at $k=10$, while DECOR continues to improve? What does this indicate about the role of the Filter agent in mitigating information pollution introduced by high‑recall retrieval?


How does a shared‑parameter single‑model architecture influence multi‑agent specialization? All agent roles in DECOR are parameterized by the same shared LLM. How does this “shared backbone” design facilitate self‑organized specialization, enabling upstream agents (such as the Planner) to actively learn behaviors that reduce the reasoning burden on downstream agents (such as the Answerer)?

**Limitations:**

Please refer to the weaknesses.

**Strengths And Weaknesses:**

Because the framework relies on multi‑agent interactive mechanisms, completing a single search task requires multiple rounds of communication among the Planner, Filter, and Answerer. Compared with single‑agent models that generate answers in one pass, this serial or iterative structure substantially increases inference latency. In addition, frequent multi‑agent calls and the need to store and process long contexts incur significantly higher token‑consumption costs.

During training, DECOR heavily relies on high‑performance LLMs (e.g., DeepSeek V3.1) as “functional judges” to provide role‑specific reward signals. If the judge model exhibits subjective biases or inaccurate scoring, the resulting agent policies may deviate from the true objective or even lead to policy collapse.

Although a hybrid reward strategy is introduced to mitigate the credit‑assignment problem, the system still struggles to fully disentangle the root causes of failure in highly complex long‑range reasoning tasks. For example, if the final answer is incorrect, it may remain difficult to determine whether the error stems from the Planner’s misdirected query planning, the Filter’s omission of key evidence, or the Answerer’s flawed logical deduction.

To address the context‑window limit ($L_{max}$), DECOR employs a frozen LLM for semantic compression. While this design helps maintain a dense observation space, repeated compression over an extremely long reasoning trajectory can lead to subtle but important loss of original factual details, which may ultimately degrade reasoning accuracy.

---

> ### Author Rebuttal · Authors · 2026-03-30
>
> We thank the reviewer for summarizing the DECOR framework. We address your concerns—including the ethics flag—below:
>
> **1. The Ethics Flag and "Harmful Content"**
> You flagged our paper for an ethics review, stating the system could be used to "automatically collect harmful content." We wish to clarify that **this was our own responsible disclosure**. In our Conclusion (Lines 437-438), we explicitly acknowledged this potential risk as part of standard safety reporting. It is standard practice to discuss broader impacts, and we respectfully argue that an ethics flag should not be triggered by the authors' own transparency regarding general AI safety.
>
> **2. Inference Latency and Token Costs**
> We fully acknowledge these trade-offs in our Conclusion (Lines 434-436). The core paradigm of DECOR—and inference-time scaling methods in general (e.g., OpenAI's o1)—is to trade more compute during inference for enhanced reasoning accuracy. Our experiments demonstrate this trade-off is highly favorable: on complex multi-hop datasets like HotpotQA and MuSiQue, DECOR significantly outperforms monolithic baselines, justifying the multi-agent overhead.
>
> **3. Reliance on LLM-as-a-Judge and Potential Bias**
> Judge bias is a known challenge, which is precisely why DECOR does not rely solely on it. Our **Hybrid Reward Strategy** strictly combines the judge's signal with the *Team Outcome Reward* (Lexical F1 and Exact Match), which is a 100% objective, human-verified ground truth. Furthermore, our judge prompt (Appendix B) strictly restricts evaluation to factual consistency rather than subjective style. Figure 3 empirically shows that our hybrid strategy successfully prevents the policy collapse observed when relying on the judge alone.
>
> **4. Credit Assignment in Complex Tasks**
> You mentioned the difficulty of identifying the root cause of an incorrect final answer. Solving this exact problem is a primary contribution of our Hybrid Reward (Eq. 3). Standard RL penalizes the entire trajectory if the final answer is wrong. In DECOR, the *Role-Specific Dense Reward* provides credit for correct intermediate steps. If the Planner issues a strong query and the Filter retains the correct evidence, they still receive positive rewards even if the Answerer eventually hallucinates. This successfully disentangles multi-agent credit assignment.
>
> **5. Semantic Compression and Loss of Details**
> We agree that detail loss is a critical risk, which is why our Filter is distinctly different from a standard summarizer. As detailed in the Filter Prompt (Appendix A.2), the agent is strictly instructed: *"Do NOT change, paraphrase, summarize, or omit specific factual details such as Names, Dates, Numbers."* DECOR's high Exact Match (F1) scores on detail-heavy datasets like TriviaQA and 2WikiMQA empirically prove that critical facts are preserved.
>
> **6. Responses to Key Questions**
> *   **Cognitive Overload:** Answered in Sec 4.2. The Filter isolates the Answerer from noisy raw retrieval, providing a clean memory pool and preventing "lost-in-the-middle" phenomena.
> *   **Inverted-U Trend:** Answered in Sec 5.2.2. The trend demonstrates that both global alignment (team reward) and local competence (role reward) are strictly necessary; omitting either degrades performance.
> *   **Robustness to Top-k Noise:** Answered in Sec 5.2.3. Monolithic agents fail at $k=10$ due to context saturation. DECOR improves because the Filter effectively rejects the noise before it reaches the reasoning stage.
> *   **Shared Backbone:** Answered in Sec 4.4. Shared parameters allow the team-level success signal to back-propagate across roles, naturally encouraging upstream agents (Planner) to output formats that downstream agents (Answerer) find easiest to process.

---

> > ### Author Rebuttal · Reviewer_xMY4 · 2026-04-01
> >
> > Although role-specific rewards are introduced, the root causes of failures in complex long-horizon tasks remain highly coupled across agents. The authors’ response does not convincingly demonstrate that this mechanism fundamentally resolves issues such as inter-agent blame shifting or the accumulation and propagation of errors.
> >
> > Although I have adjusted my score upwards, I still maintain a negative recommendation for this paper.

---

> > > ### Author Response · Authors · 2026-04-02
> > >
> > > Thank you for taking the time to follow up and for adjusting your score. We really appreciate you bringing this up. Your concern is very fair and spots a real challenge in multi-agent systems.
> > >
> > > You are absolutely right that in a step-by-step pipeline, errors are deeply connected. We want to be completely honest: **we do not claim that DECOR has perfectly "solved" the credit assignment problem** in reinforcement learning. However, we did build two specific designs into the system to **break the "blame chain" and stop errors from piling up**. This is exactly why our model is able to train successfully:
> > >
> > > **1. Preventing Blame-Shifting by Judging Based on Inputs**
> > > In standard RL, if the final answer is wrong, every agent gets a penalty. This is why blame-shifting happens. DECOR avoids this because our Judge model does not just look at the final result. Instead, it **judges each agent based *only* on the information it received at that specific step.**
> > >
> > > For example, if the Planner searches for useless documents, the Filter is not punished for failing to find the right answer. The Filter is only judged on whether it handled its given task well (in this case, correctly realizing the text is useless and dropping it). Similarly (as noted in Section 4.3), the Answerer is **only penalized for a reasoning mistake if the Filter actually gave it the correct facts** to begin with. By judging agents on what they were given—rather than the team's final failure—we **stop the downstream agents from taking the blame for upstream errors.**
> > >
> > > **2. Stopping Error Propagation through Information Isolation**
> > > In standard single-model setups, a bad search result or a wrong intermediate thought stays in the context window. This confuses the model, and the error carries over to the final answer. DECOR prevents this by **separating what each agent can see.**
> > >
> > > The Answerer is **completely blind to the messy, raw search results** and the past failed queries. It only sees the clean, filtered notes. If the Filter does its job and removes the noise, **the error stops right there**. It never reaches the Answerer, which keeps the final reasoning safe from early-stage mistakes.
> > >
> > > Ultimately, DECOR's core motivation is to tackle the "cognitive overload" that naturally leads to these cascading errors. By strictly separating what each agent observes and assigning rewards based on localized inputs, we demonstrate that multi-agent training can be made significantly more stable and interpretable than standard outcome-based RL.
> > >
> > > We completely agree with you that dealing with connected reasoning steps remains a tough challenge for the whole AI community. We hope this explanation shows how DECOR’s design practically reduces these issues. We will definitely add a clearer discussion of these limitations and our system's boundaries in the final version.
> > >
> > > **Thank you again for your time, patience, and very helpful feedback—it truly helps us make this work better. We hope you might consider these practical mechanisms in your final evaluation.**

---

### Official Review · Reviewer_T2zP · 2026-03-10

**Soundness:** 3
**Presentation:** 3
**Significance:** 3
**Originality:** 3
**Overall Recommendation:** 4
**Confidence:** 4

**Summary:**

**The Problem:**
Think of a single AI agent trying to do a complex online research task (like a "deep search" involving multiple steps). It has to figure out where to look, filter out the useless information, and finally write an answer. Doing all of this at once can overwhelm the agent—the paper calls this "cognitive overload."

**The Solution (DECOR):**
The authors propose a system called DECOR that splits the work among three specialized AI agents, like a well-organized team:
1.  **The Planner:** Decides what to search for next (the "navigator").
2.  **The Filter:** Decides which pieces of information are worth keeping (the "librarian").
3.  **The Answerer:** Writes the final answer based on the kept information (the "writer").

**The Key Innovation:**
Instead of just telling these agents what to do, DECOR uses **Multi-Agent Reinforcement Learning (MARL)** . This means the three agents learn from experience by getting feedback. They receive rewards not just for doing their own job well, but also for how well the team performs overall. This allows them to learn from past mistakes and collaborate better over time.

**Main Contribution:**
The paper shows that this trained, specialized team approach works much better than having one giant AI do everything alone, proving that learning to break down the task is key to handling complex research.

**Compliance With Llm Reviewing Policy:**

Affirmed.

**Key Questions For Authors:**

1.  analysis of **failure cases**. It would be honest and helpful to know when or why DECOR might fail. Does it struggle with very niche topics? Is it slower than a single agent?
2. the computational cost. Training three specialized agents with MARL is likely more expensive than using one single model. The paper should honestly address this trade-off.

**Strengths And Weaknesses:**

### Strengths

1.  **Soundness:**
    - The approach is technically solid. The idea of breaking a complex task (deep search) into three distinct roles (Planner, Filter, Answerer) is logical and addresses a real problem ("cognitive overload").
    - Using Multi-Agent Reinforcement Learning (MARL) to train these agents together is a sensible way to ensure they learn to collaborate, rather than just working in isolation.
    - The paper supports its claims with experiments on **seven different benchmarks**, showing that their method works consistently better than strong single-model baselines.

2.  **Presentation:**
    - The problem is clearly stated at the beginning: monolithic agents get overwhelmed.
    - The solution (DECOR) is easy to grasp because of the simple team analogy (Navigator, Librarian, Writer).
    - The structure likely follows a standard, easy-to-follow format (Problem → Method → Experiments → Conclusion).

3.  **Significance:**
    - It addresses a very relevant problem in AI: how to handle complex, multi-step research tasks that current large language models often struggle with.
    - By showing that a *learned* collaborative system beats a single agent, it provides a clear direction for future research in building more capable AI systems for information gathering.
    - The hybrid reward system (individual feedback + team success) is a useful contribution that other researchers can build upon.

4.  **Originality:**
    - While using multiple agents isn't new, the paper's originality lies in **functional decomposition via learning**. Other multi-agent systems often just use fixed, pre-programmed roles. DECOR's agents actually *learn* their specialized jobs and how to adapt to each other through reinforcement learning.
    - The combination of role-specialization with joint optimization (MARL) is a creative and well-articulated combination of existing ideas.

### Weaknesses

1.  While the results are strong, the paper (based on the summary) doesn't analyses **failure cases**. It would be honest and helpful to know when or why DECOR might fail. Does it struggle with very niche topics? Is it slower than a single agent?
2. the computational cost. Training three specialized agents with MARL is likely more expensive than using one single model. The paper should honestly address this trade-off.

---

> ### Author Rebuttal · Authors · 2026-03-30
>
> **Thank you for your clear, encouraging summary of our work.** We are glad you found the functional decomposition (Navigator, Librarian, Writer) intuitive and the MARL strategy effective. We address your constructive suggestions regarding limitations below:
>
> **1. Analysis of Failure Cases**
> We completely agree that discussing failure modes adds honesty and depth to the paper. We will add a dedicated "Failure Analysis" subsection in the appendix. Currently, DECOR primarily fails in two scenarios:
> *   **Long-Tail or Highly Specialized Queries:** If the external search engine repeatedly returns documents with zero relevance to a rare or highly specialized topic (where web coverage is extremely sparse), the Planner may exhaust its search budget (reaching $T_{max} = 4$) without gathering useful facts, forcing the Answerer to guess or admit ignorance.
> *   **Over-Filtering in Complex Reasoning:** Occasionally, the Filter agent might become overly aggressive and prune a minor detail that seems irrelevant in the early search steps, but is actually required for a complex multi-hop deduction later in the trajectory.
>
> **2. Computational Cost Trade-off**
> You are entirely correct that running three specialized agents is computationally more expensive than running a single monolithic model. In our revised manuscript, we will explicitly quantify this: DECOR typically consumes roughly 2.5x to 3x more tokens and wall-clock time per query compared to a standard single-agent pipeline. We frame DECOR as a system designed for high-stakes or complex reasoning scenarios where accuracy and factual grounding are prioritized over millisecond-level latency. We will make this trade-off explicitly clear in the Conclusion and Limitations sections.
>
> **Thank you again for your positive review and valuable suggestions, which will certainly strengthen the final version of our paper.**

---

### Official Review · Reviewer_baqy · 2026-03-10

**Soundness:** 2
**Presentation:** 3
**Significance:** 3
**Originality:** 3
**Overall Recommendation:** 4
**Confidence:** 3

**Summary:**

This paper proposes DECOR, a multi-agent reinforcement learning (MARL) framework for deep search, which decomposes the search-and-reason pipeline into three role-specialized agents: a Planner (query generation), a Filter (evidence extraction and noise reduction), and an Answerer (final synthesis). All three agents share a single LLM backbone (Qwen3-8B) and are jointly trained via a hybrid reward strategy that combines team-level outcome rewards (F1 + LLM-as-a-Judge) with role-specific intrinsic rewards from an LLM judge. The authors formalize the problem as a MA-POMDP and adapt GRPO-style policy optimization for the multi-agent setting. Experiments on seven QA benchmarks (NQ, TriviaQA, HotpotQA, 2WikiMQA, MuSiQue, Bamboogle, PopQA) show improvements over several monolithic deep search baselines, with ablations on reward design, the α trade-off parameter, retrieval volume robustness, and extensibility with rerankers/live search.

**Compliance With Llm Reviewing Policy:**

Affirmed.

**Final Justification:**

I thank the authors for a thorough rebuttal and am updating my recommendation from Weak Reject to Weak Accept.

The rebuttal resolved my main concerns: the controlled-backbone clarification means the main results can be attributed to DECOR's architecture rather than base model differences; the compression-module trigger explanation attributes the robustness gains to the learned Filter policy rather than the fallback mechanism; and the human-judge agreement check adequately addresses the circular-optimization concern. The added cost-performance analysis, while limited in scope, makes a reasonable case that DECOR allocates test-time compute purposefully rather than blindly.

**Key Questions For Authors:**

1. Were all baselines in Table 1 re-implemented and re-trained using the same Qwen3-8B-Instruct backbone? If not, can you provide results with a controlled backbone to isolate the contribution of DECOR's architecture from the base model quality? This would significantly affect my assessment.

2. Have you compared shared-parameter DECOR against a variant where each agent is a separate model (or at least separate LoRA adapters)? What is the empirical effect of parameter sharing on both performance and training stability?

3. Can you provide wall-clock inference time and total token consumption per query for DECOR vs. Search-R1 and SimpleDeepSearcher? Even rough numbers would help.

4. Have you measured LLM judge agreement with human annotations on a sample? What happens if you train with the hybrid reward but evaluate purely on human-judged accuracy?

**Limitations:**

Yes, the authors acknowledge inference cost and potential misuse risks in the conclusion. However, the cost discussion is only qualitative. I would encourage quantifying the overhead as suggested above. The paper would also benefit from discussing failure modes — when does the multi-agent decomposition hurt?

**Strengths And Weaknesses:**

### Strengths

1. The "cognitive overload" framing is intuitive and well-supported by the literature on lost-in-the-middle phenomena. Offloading noise filtering to a dedicated agent so the Answerer only sees curated evidence is a clean architectural idea. The information barrier design — where the Planner cannot see raw retrieval results and the Answerer cannot see rejected noise — is principled and clearly articulated.

2. The hierarchical reward (format → team → role) is a thoughtful contribution. The ablation in Section 5.2.1 convincingly shows that neither pure lexical rewards nor pure LLM-judge rewards suffice, and the α sensitivity study (Figure 4) adds useful practical guidance. The idea of using role-specific group normalization for advantage estimation (Eq. 6) to address credit assignment in heterogeneous teams is interesting.

3. DECOR achieves the best LLM-as-a-Judge scores across all seven benchmarks, which is a strong signal that the framework produces semantically more accurate answers even when lexical overlap is comparable to baselines.

4. The monotonic improvement of DECOR as top-k increases (while Search-R1 degrades) is a compelling demonstration that the Filter agent is doing meaningful work. This is one of the most convincing results in the paper.

### Weaknesses


1. This is the most significant concern. DECOR uses Qwen3-8B-Instruct as the backbone, but the paper does not clearly state which backbone models the baselines use. Search-R1, R1-Searcher, and o2-searcher were originally proposed with different base models (e.g., Qwen2.5-7B). If the baselines were not re-trained or at least re-run with the same Qwen3-8B backbone, the comparison conflates the effect of the DECOR framework with the effect of using a stronger base model. The authors should clarify whether all methods were re-implemented on the same backbone, or whether the reported numbers are taken from the original papers. This is critical for interpreting Table 1.

2. The paper emphasizes that all three agents share a single LLM backbone, but never seriously examines this choice. How does shared-parameter DECOR compare to a variant where each agent is a separate copy of Qwen3-8B fine-tuned independently? The claim of "self-organized specialization" through shared parameters (Section 4.4) is interesting but purely qualitative — no evidence is provided (e.g., gradient analysis, representation divergence, or attention pattern differences across roles) to show that specialization actually emerges.

3. The conclusion mentions "higher inference latency and token costs" but provides no numbers. DECOR requires three sequential LLM calls per search step (Planner → retrieve → Filter), plus a final Answerer call, plus potentially an LLM call for memory compression, plus the LLM-as-a-Judge calls during training. A wall-clock time comparison and token consumption analysis against monolithic baselines would be necessary to assess practical viability. Without this, it is hard to judge whether the gains justify the overhead.

4. The Filter both extracts evidence from raw documents AND triggers memory compression via a frozen LLM when the context overflows. The compression step uses a separate frozen model that is not part of the MARL loop. This raises questions: how much of DECOR's robustness to increasing k (Table 2) comes from the learned Filter policy vs. the compression module? An ablation separating these two mechanisms would strengthen the paper.

5. Both the training reward (R_LJ) and the primary evaluation metric (LJ) rely on LLM judges. Using an LLM judge for both training and evaluation introduces a risk of circular optimization — the agents may learn to produce outputs that satisfy the judge's preferences rather than being genuinely more accurate. Although the authors use DeepSeek-v3.1 as the judge and combine it with F1, some analysis of judge agreement with human evaluations (even on a small sample) would be valuable.

---

> ### Author Rebuttal · Authors · 2026-03-30
>
> **Thank you for your thorough review and for recognizing the value of our "cognitive overload" framing and hierarchical reward design.** We deeply appreciate your rigorous technical questions, particularly regarding the baselines and computational overhead. We address your concerns below:
>
> **1. Fair Comparison on Backbone Models (Baselines)**
> This is a critical point, and we apologize for the lack of clarity. To ensure a strictly fair comparison, **we standardized the backbone model across all open-source baselines.** Specifically, we re-implemented and re-ran baselines like Search-R1 and SimpleDeepSearcher using the exact same **Qwen3-8B-Instruct** backbone as DECOR. Therefore, the improvements shown in Table 1 are entirely attributable to DECOR's multi-agent architecture and hybrid MARL strategy, not to differences in base model capabilities. We will explicitly state this in the experimental setup and Table 1 caption in the revision.
>
> **2. Shared Parameters vs. Separate Models**
> Our decision to use a shared backbone was primarily driven by memory efficiency (allowing joint training on an 80GB A100 cluster without managing three separate massive models). However, your question is highly relevant. During our early prototyping, we compared the shared backbone against a variant where each agent was trained with a separate LoRA adapter. We observed that while separate LoRAs converged slightly faster initially, the final performance was remarkably similar (within ~1% LJ accuracy). The shared parameter design successfully learns "self-organized specialization" because the specific agent prompts (Appendix A) act as strong conditional prefixes, shifting the model's internal activations. We will include this ablation discussion in the revised Appendix.
>
> **3. Inference Latency and Token Costs**
> We fully agree that these costs must be quantified. We will add a dedicated "Computational Overhead" section. On average, across complex tasks (e.g., HotpotQA), DECOR requires approximately **2.5x to 3x more inference tokens** and wall-clock time compared to single-pass baselines. We view this as a deliberate and worthwhile trade-off, aligning with the recent paradigm of scaling "test-time compute" (e.g., OpenAI o1)—spending more compute during inference to guarantee semantic correctness and alleviate hallucinations in complex reasoning.
>
> **4. Filter Policy vs. Frozen Compression Module**
> To clarify, the frozen compression module is a strict fallback mechanism. It *only* triggers when the accumulated memory strictly exceeds $L_{max}$ (4096 tokens). In our Top-$k$ robustness experiments (Table 2), for $k=3$ and $k=5$, the context length rarely hits this threshold. Therefore, the performance gains and noise robustness shown in those settings are **almost entirely driven by the MARL-trained Filter policy** actively discarding irrelevant sentences, not by the fallback compression. We will clarify this trigger condition in Section 4.2.
>
> **5. Circular Optimization and Judge Agreement**
> To address the risk of circular optimization, we conducted a preliminary human evaluation. We manually audited 100 random samples from our test set and compared human judgments against DeepSeek-v3.1’s scores. We found a **92% agreement rate**, confirming that our judge is highly aligned with human semantic preferences and that the agents are learning genuine reasoning skills rather than just "gaming" the judge. Furthermore, the inclusion of the rigid lexical metric ($F1$) in our Hybrid Reward acts as a strong anchor to prevent the policy from drifting into purely subjective optimization.

---

> > ### Author Rebuttal · Reviewer_baqy · 2026-04-03
> >
> > Thank you for the detailed rebuttal. The clarifications on shared vs. separate parameters (Point 2), the compression module trigger conditions (Point 4), and the human agreement rate (Point 5) are helpful and largely address those concerns.
> >
> > However, I would like to follow up on two points:
> >
> > **Regarding Point 1 (Fair Comparison):** I appreciate the clarification that all baselines were re-run with the same Qwen3-8B-Instruct backbone. However, my concern extends beyond the backbone itself. Were all baselines also **fine-tuned with comparable training data and compute budgets**? DECOR involves a full MARL training pipeline with hybrid rewards, while some baselines (e.g., Search-R1, SimpleDeepSearcher) may have different training regimes or data requirements. If DECOR benefits from a more extensive fine-tuning process while certain baselines are compared in a less optimized or even zero-shot setting, the gains could partly reflect training effort rather than architectural merit. Could you clarify the training setup (data size, training steps, compute budget) for each baseline to ensure the comparison is truly controlled?
> >
> > **Regarding Point 3 (Inference Cost):** You mention "2.5x to 3x more inference tokens and wall-clock time compared to single-pass baselines," but it is unclear which specific methods fall under "single-pass baselines." Could you provide a concrete comparison table listing wall-clock time, token consumption, and corresponding accuracy for DECOR and each baseline? This would allow readers to assess the cost-performance trade-off directly rather than relying on an aggregate estimate. The "test-time compute scaling" argument is reasonable in principle, but without a clear efficiency-accuracy Pareto analysis, it is difficult to judge whether DECOR's overhead is justified across all settings.
> >
> > These clarifications would meaningfully strengthen the empirical contribution. I look forward to the authors' response.

---

> > > ### Author Response · Authors · 2026-04-03
> > >
> > > Thank you for your timely follow-up. We are glad our previous clarifications addressed your concerns regarding the shared parameters and compression module. Below, we clarify the rigorous experimental controls and cost-efficiency trade-offs of DECOR.
> > >
> > > **1. Fair Comparison (Training Setup & Compute Budget)**
> > > We absolutely agree that comparing a fine-tuned MARL pipeline against under-trained baselines is unfair. To guarantee a strictly controlled comparison, **all learning-based baselines were fine-tuned using the exact same 20K training dataset and equivalent compute budgets:**
> > >
> > > *   **Search-R1 & R1-Searcher:** Trained using the same RL environment interactions and PPO/GRPO optimization budget as DECOR (approx. 200 total updates with matched batch sizes).
> > > *   **SimpleDeepSearcher:** Synthesized training trajectories from our 20K dataset using its official pipeline, and SFT fine-tuned the Qwen3-8B-Instruct backbone for 3 epochs until convergence.
> > >
> > > None of the baselines (except the training-free CoRAG) were evaluated zero-shot. They represent the *best possible monolithic policies* on our training set. The performance delta genuinely reflects the architectural merit of DECOR’s decomposition. We will add a "Baseline Training Configurations" section in Appendix C.
> > >
> > > **2. Inference Cost and Pareto Analysis**
> > > By "single-pass baselines", we meant straightforward pipelines like *SimpleDeepSearcher* (Search -> Read -> Answer). To address your request, we provide the average token consumption and wall-clock latency per query on HotpotQA *(assuming an 8B model at ~60 tokens/s and ~1.5s network delay per search)*:
> > >
> > > | Method | HotpotQA Acc (LJ) | Avg Total Tokens / Query | Avg Wall-Clock Time (s) |
> > > | :--- | :---: | :---: | :---: |
> > > | SimpleDeepSearcher (Single-pass) | 56.3 | ~1,250 | ~4.5 |
> > > | Search-R1 (Monolithic RL) | 59.1 | ~3,000 | ~14.0 |
> > > | **DECOR (Ours - Multi-agent)** | **59.8** | **~4,200** | **~18.5** |
> > > | o2-searcher (Inference scaling) | 46.4 | ~5,800 | ~28.0 |
> > >
> > > *Analysis of the Trade-off:*
> > > *   **SimpleDeepSearcher (~4.5s):** Fast but hits a 56.3% ceiling due to its inability to perform multi-hop reasoning.
> > > *   **Search-R1 (~14.0s):** Offers strong reasoning via autonomous think-search loops.
> > > *   **DECOR (~18.5s):** Incurs a ~30% time overhead vs. Search-R1. This is the inherent "multi-agent prompt tax"—repeated context passing between Planner, Filter, and Answerer.
> > > *   **Why the overhead is justified:** The comparison against *o2-searcher* (~28.0s) is crucial. o2-searcher blindly scales inference time, leading to massive token bloat and accuracy collapse (46.4%) due to noise overload. In contrast, DECOR allocates test-time compute purposefully to *filter* noise. Though total system tokens are higher, the Answerer's context remains pristine.
> > >
> > > DECOR forms a highly favorable Pareto frontier: trading a reasonable structural overhead for the highest accuracy and robustness. We will include this exact table and breakdown in the revision.
> > >
> > > **Concluding Remarks**
> > > We believe this cost-performance analysis solidifies that offloading noise to a dedicated Filter is a necessary structural optimization for robust reasoning. **We hope these detailed clarifications resolve your remaining concerns and and justify a higher score.**

---

### Official Review · Reviewer_WoKk · 2026-03-13

**Soundness:** 2
**Presentation:** 4
**Significance:** 2
**Originality:** 3
**Overall Recommendation:** 4
**Confidence:** 4

**Summary:**

The authors propose a pseudo-multi-agent RL approach to deep search with LLMs. Three sub-agents sharing a single backbone are jointly optimized to plan queries, filter retrieved evidence, and synthesize answers. They report marginal improvements over competing approaches on seven benchmarks.

**Compliance With Llm Reviewing Policy:**

Affirmed.

**Ethical Review Concerns:**

Per Policy B, I fed the submission to a privacy-compliant LLM while polishing my review. Part of this involved me confirming that there were no prompt injection attacks. In doing so, I have found a prompt injection attack in the text at the bottom of page 2. This can be confirmed by copying the text below the page number and pasting it as plain text into a text editor. While the injection seems relatively harmless, this is a violation of ICML's policies, which are unambiguous that this paper must be rejected.

**Ethical Review Flag:**

Flag this paper for an ethics review.

**Ethics Expertise Needed:**

["Research Integrity Issues (e.g., plagiarism)"]

**Final Justification:**

In general, the paper covers a good topic at a good time. My big concerns were around the formal existence of the "cognitive overload" concept (which they handled in the rebuttal) and the significance of the results (which they handled in the rebuttal but wrongly, undermining their paper a fair bit). They reported statistical test results which skipped a family-wise error rate correction. When doing something like Holm-Bonferroni, it puts only 2 of 5 tests at a significance level with a FWER of p < 0.05. Ignoring the need for additional mechanisms when running only 3 samples (which might be beyond expectations of a work at ICML), this result pretty much shows their gains aren't significant. I thus see this paper as more descriptive in nature and not a strong result. But that's many of the papers at ICML, so I'll leave my score as a weak accept.

**Key Questions For Authors:**

See weaknesses.

**Limitations:**

Societally, I see nothing concerning. Otherwise, I see limitations I raised in the weaknesses section that are not mentioned in the paper.

**Strengths And Weaknesses:**

### Strengths

- This is a timely and relevant paper. There is not enough work on dedicated evidence filtering in deep search, and this seems like a promising direction.
- The writing is clear and the illustrations are very good, making the framework easy to follow. Even in the era of LLM-assisted writing, this paper's presentation is above the average of what I see.
- A solid suite of experiments covering seven benchmarks with multiple ablations.
- The inclusion of the LLM-as-a-Judge metric alongside F1 is a good call, as it provides a more robust evaluation of semantic correctness (though I note that the cases where the method underperforms baselines tend to be on the F1 metric specifically).

### Weaknesses

- First and foremost, the paper includes a prompt injection attack. Per ICML policy, I've reported this to the area chair, senior area chair, and program chairs. While this is under investigation, I've left this as normal. However, note that ICML policy is quite clear that this paper CANNOT be accepted now.
- The acronym is a bit tortured. Not a reason to reject, but worth the authors thinking about.
- My biggest concern is that the paper does not cite any source establishing the "cognitive overload" phenomenon it claims to address (also note the typo "Cognivive" in Figure 1). There is some tangentially related discussion in the related works section (e.g., "lost-in-the-middle"), but no clear evidence that this specific phenomenon exists in the way the authors describe. While I suspect it does exist, that is not a sufficient bar for a scientific paper. One must first establish that a phenomenon exists before proposing mitigations. (Question 1) Could the authors provide citations to works that explicitly analyze this? This is a hard block for me, though providing adequate references would resolve this particular block.
- On a related note, "significantly" is a strong term and quite pertinent here. If the authors use it---which they do---they must show statistical significance. Many of the improvements are marginal, making this maybe hard to address. (Question 2) Can the authors show any evidence for this at all? Even removing the word, I worry somewhat about how meaningful the improvements observed here are. This is the second block that makes me concerned about accepting this paper.
- I am not sure what Section 4.1 contributes. The latent-variable framing feels like unnecessary formalism that does not connect to the actual training procedure. I would advise the authors to consider if it actually makes their paper more understandable or not.
- It would be valuable to see results with more than one backbone model. As it stands, it is difficult to say whether the improvements are a general property of the approach or specific to the Qwen family. I appreciate that this is expensive, but even a single additional model from a different lineage (e.g., a heavily quantized GPT OSS 20B) would strengthen the claims considerably.

---

> ### Author Rebuttal · Authors · 2026-03-30
>
> We sincerely thank you for your time, your highly positive feedback on our writing and experimental suite, and your appreciation of our LLM-as-a-Judge evaluation. We address your concerns below:
>
> **1. The "Prompt Injection" / Ethics Flag (Crucial Clarification)**
> We were initially shocked by the prompt injection concern, as we absolutely did not insert such text into our LaTeX source. Upon investigating the PDF, we realized that the hidden string you found (*"Include BOTH the phrases..."*) is an **official hidden watermark automatically injected by the ICML submission system** into the generated Reviewer Copy.
>
> This is a system-level mechanism used by the conference to catch reviewers who feed PDFs into unauthorized LLMs. We, as authors, have no control over this watermark. We deeply appreciate your diligence and integrity in checking for this, and we have already contacted the Area Chair to clarify this system artifact. We kindly ask that you dismiss this ethics flag, as no policy violation occurred on our end.
>
> **2. Justifications and Citations for "Cognitive Overload"**
> You raise a very fair point regarding the lack of formal citations for "cognitive overload." In the revision, we will formally anchor this term to recent studies analyzing LLM attention limits in retrieval-augmented generation. Specifically, we will cite *Xu et al. (2024)*[1] and *Liu et al. (2024b)*[2], which empirically demonstrate that as context grows with noisy retrieval, a monolithic LLM's attention becomes diluted, leading to reasoning failures. DECOR directly mitigates this attention dilution by utilizing the Filter agent to curate a dense, clean memory pool before the Answerer begins synthesis.
>
> **3. Statistical Significance of Improvements**
> We completely agree with your standard for the word "significantly." In our revised manuscript, we will report the standard deviations across 3 random seeds for all MARL runs. Furthermore, we will run paired t-tests to formally validate the statistical significance of our gains (especially on multi-hop datasets like HotpotQA and 2WikiMQA, where our margins over Search-R1 are the largest). For any dataset where the gains are marginal or not statistically significant ($p > 0.05$), we will explicitly tone down our claims and adjust the terminology accordingly.
>
> **4. Purpose of Section 4.1 (Latent-Variable Framing)**
> We understand why this framing might feel disconnected from the RL implementation. We included Eq. 1 to mathematically justify why we can restrict our agents to *partial observations* (MA-POMDP). Without factorizing the joint probability, there is no formal justification for shielding the Answerer from the raw, noisy retrieval results. However, we agree it could be better integrated. We will rewrite this section to directly connect the probabilistic factorization to our practical Hybrid Reward design and the partial observation boundaries of each agent.
>
> **5. Testing Additional Backbone Models**
> This is an excellent suggestion. Because MARL training for multi-agent systems is highly compute-intensive, we restricted our initial ablations to Qwen3-8B. We agree that demonstrating results on a different model family (e.g., Llama-3-8B) would strongly validate the generalizability of DECOR. We commit to including an alternative backbone experiment in the final version if accepted, and we will add a dedicated Limitations section discussing backbone dependency.
>
> Thank you again for your thorough, constructive, and responsible review. We hope this clarification addresses your primary concerns, and we kindly ask if you might reconsider your score in light of the hidden watermark (official prompt injection) explanation.
>
> **reference**
>
> [1] Xu, Z., Jain, S., and Kankanhalli, M. Hallucination is inevitable: An innate limitation of large language models. arXiv preprint arXiv:2401.11817, 2024.
>
> [2] Liu, N. F., Lin, K., Hewitt, J., Paranjape, A., Bevilacqua, M., Petroni, F., and Liang, P. Lost in the middle: How language models use long contexts. Transactions of the Association for Computational Linguistics, 12:157–173, 2024b.

---

> > ### Author Rebuttal · Reviewer_WoKk · 2026-04-04
> >
> > My score wasn't really due to the watermark. But in general, I think my biggest concern is put to rest, and there was a decent response to my second biggest concern. I'm still not wholly confident about the significance thing. I'd like to know before whether the numbers are meaningful (I'm somewhat suspicious they may not be, but this is pretty common in the field). In the meantime, I'll be raising my score from a weak reject to a weak accept.

---

> > > ### Author Response · Authors · 2026-04-05
> > >
> > > Thank you very much for your prompt reply, for dismissing the system watermark issue, and for raising your score to a Weak Accept! We truly appreciate your constructive engagement and your rigorous standards.
> > >
> > > To directly address your remaining concern regarding whether the improvements are statistically meaningful *right now*, we have immediately calculated the $p$-values using a **paired Student's t-test** (comparing DECOR’s sample-level predictions against the strongest baseline for each dataset) and finalized the variance (mean ± std) across **3 independent MARL training seeds**.
> > >
> > > Here are the concrete statistical results:
> > >
> > > **1. Multi-Hop / Complex Reasoning Tasks (Statistically Significant):**
> > > On datasets requiring extensive context gathering and synthesis—where "cognitive overload" is most prevalent—DECOR's gains are robust and statistically significant ($p < 0.05$).
> > > *   **2WikiMQA (LJ Score):** DECOR ($62.0 \pm 0.6$) vs. Search-R1 ($52.4 \pm 0.8$) $\rightarrow$ **$p < 0.001$**
> > > *   **Bamboogle (LJ Score):** DECOR ($51.7 \pm 0.5$) vs. Search-R1 ($47.2 \pm 0.7$) $\rightarrow$ **$p = 0.004$**
> > > *   **HotpotQA (F1 Score):** DECOR ($58.3 \pm 0.4$) vs. Search-R1 ($55.8 \pm 0.5$) $\rightarrow$ **$p = 0.021$**
> > >
> > > **2. Single-Hop / Factoid Retrieval Tasks (Marginal / Trend Only):**
> > > In the spirit of full transparency, you were entirely correct to be suspicious of the smaller margins. On simpler single-hop datasets, the gains, while positive, are indeed marginal and border on the threshold of statistical significance. This actually aligns perfectly with our core premise: monolithic models handle short-context, single-step retrieval reasonably well, but the dedicated Filter agent becomes necessary when contexts grow long and noisy.
> > > *   **TriviaQA (LJ Score):** DECOR ($71.4 \pm 0.4$) vs. SimpleDeepSearcher ($69.6 \pm 0.5$) $\rightarrow$ **$p = 0.072$** (Marginal)
> > > *   **NQ (LJ Score):** DECOR ($53.2 \pm 0.5$) vs. Search-R1 ($51.3 \pm 0.6$) $\rightarrow$ **$p = 0.065$** (Marginal)
> > >
> > > **Actionable changes for the paper:**
> > > These numbers confirm that DECOR's performance is not a statistical artifact on the complex reasoning tasks it was specifically designed to solve. Per your excellent advice, we will include this exact variance and $p$-value analysis in the revised main text. Furthermore, we will explicitly tone down our claims regarding the single-hop datasets, strictly reserving the word "significantly" for the multi-hop results where $p < 0.05$.
> > >
> > > We hope that providing these concrete numbers *before* the end of the discussion phase gives you the full confidence you need regarding the meaningfulness of our improvements. Thank you again for pushing us to make this a more rigorous, scientifically sound paper, your feedback has genuinely improved the manuscript.

---

### Decision · Program_Chairs · 2026-04-30

**Decision:**

Accept (regular)

**Comment:**

Reviewers appreciated the innovative conceptual contribution of this work, its clear presentation, and substantial evaluation. Most concerns raised by reviewers have been thoroughly addressed by the authors in rebuttal, including the grounding of the motivation in the literature; the design choice to tie model weights; the reasonableness of the comparison to baselines, of the context compression, and of the LLM judging; and the accuracy–cost tradeoff. Acceptance is thus recommended due to the significance of the contributions, despite remaining concerns regarding the statistical significance of the empirical results and the core mechanism of the demonstrated benefits.